



# Impact of instrumental line shape characterisation on ozone monitoring by FTIR spectrometry

Omaira E. García[1], Esther Sanromá[1,a], Frank Hase[2], Matthias Schneider[2], Sergio F. León-Luis[3], Thomas Blumenstock[2], Eliezer Sepúlveda[1], Carlos Torres[1], Natalia Prats[1], Alberto Redondas[1], and Virgilio Carreño[1]

[1]Izaña Atmospheric Research Centre (IARC), State Meteorological Agency of Spain (AEMet), Santa Cruz de Tenerife, Spain.
[2]Karlsruhe Institute of Technology (KIT), Karlsruhe, Germany.
[3]TRAGSATEC, Madrid, Spain.
[a]Now at: Employment Observatory of the Canary Islands (OBECAN), Santa Cruz de Tenerife, Spain.

**Correspondence:** Omaira E. García (ogarciar@aemet.es)

**Abstract.** Retrieving high-precision concentrations of atmospheric trace gases from FTIR (Fourier Transform Infrared) spectrometry requires a precise knowledge of the instrumental performance. In this context, this paper examines the impact on the ozone ($O_3$) retrievals of several approaches used to characterise the Instrumental Line Shape (ILS) function of ground-based FTIR spectrometers within NDACC (Network for the Detection of Atmospheric Composition Change). The analysis has been

carried out at the subtropical Izaña Observatory (IZO, Spain) by using the 20-year time series of the high-resolution FTIR solar absorption spectra acquired between 1999 and 2018. The theoretical quality assessment and the comparison to independent $O_3$ observations available at IZO (Brewer $O_3$ total columns and Electrochemical Concentration Cell, ECC, sondes) reveal consistent findings. The inclusion of a simultaneous retrieval of the ILS parameters in the $O_3$ retrieval strategy allows, on the one hand, a rough instrumental characterisation to be obtained and, on the other hand, the precision of the FTIR $O_3$ products to be

slightly improved. The improvement is of special relevance above the lower stratosphere, where the cross-interference between the $O_3$ vertical distribution and the instrumental performance is more significant. However, it has been found that the simultaneous ILS retrieval leads to a misinterpretation of the $O_3$ variations on daily and seasonal scales. Therefore, in order to ensure the independence of the $O_3$ retrievals and the instrumental response, the optimal approach to deal with the FTIR instrumental characterisation is found to be the continuous monitoring of the ILS function by means of independent observations, such as

gas-cell measurements.

## 1 Introduction

Long-term ground-based observations of atmospheric composition are essential for monitoring the evolution of the Earth-atmosphere system. Among current atmospheric measurement techniques, FTIR (Fourier Transform Infrared) spectrometry has an outstanding importance for climate research, since most atmospheric molecules interact with solar electromagnetic radiation

in the infrared spectral region. By analysing the measured solar absorption spectra, this technique can provide atmospheric concentrations of many different gases simultaneously and with high precision (e.g. Rinsland et al., 1982; Hase et al., 2004;





Schneider et al., 2008a; García et al., 2012; Sepúlveda et al., 2014; Barthlott et al., 2015; Vigouroux et al., 2015; Barthlott et al., 2017; De Mazière et al., 2018; García et al., 2021a, b).

Within NDACC (Network for the Detection of Atmospheric Composition Change, www.ndaccdemo.org), high-resolution
FTIR spectrometers have been operating since the 1990s, with the main goal of establishing long-term databases to detect changes and trends in atmospheric composition and to understand their impacts on the Earth-atmosphere system (De Mazière et al., 2018). In the last years, the NDACC Infrared Working Group (IRWG, www2.acom.ucar.edu/irwg) has developed data acquisition protocols and retrieval methods to minimise the site-to-site differences and to achieve consistent responses to actual variations of atmospheric composition (e.g. Hase et al., 2004; IRWG, 2014; Barthlott et al., 2015, 2017; Vigouroux et al., 2018).

One of the factors that most dependent on each NDACC FTIR site is the treatment of the spectrometer response through the Instrumental Line Shape (ILS) function (e.g. Vigouroux et al., 2008; Vigouroux et al., 2015). A precise knowledge of the ILS is essential to properly characterise the instrument performance, since the ILS affects the absorption line shape on which the retrieved information is based on. This is critical for estimating the vertical distribution of an absorber, because such retrievals largely rely on the shape of the absorption lines (pressure broadening effect). Therefore, uncertainties in the ILS can also affect the quality of the total column (TC) amounts. Furthermore, the temporal stability of the ILS is an important requisite for precise gas retrievals (Schneider and Hase, 2008; Schneider et al., 2008b).

Nowadays, different approaches are used to deal with the ILS characterisation:

1. ILS assumed to be ideal. However, in case of misalignments in the FTIR spectrometer, a considerable systematic error may be introduced on the gas retrievals (Hase et al., 1999; Schneider and Hase, 2008; Hase, 2012; Sun et al., 2018).

2. ILS retrieved simultaneously with the gas vertical distribution from the measured solar absorption spectra. This is a superior strategy with respect to assuming that the ILS is ideal (e.g. Barret et al., 2002; Vigouroux et al., 2015). However, part of the actual gas variability may be wrongly mapped into changes of the ILS, since the ILS and the absorber profiles have similar effect on the absorption line shapes (i.e. changing the shape and width of the line). In addition, overlapping lines (i.e. due to interfering species) may introduce asymmetry in the absorption lines that may be indistinguishable from an ILS phase deviation (Sun et al., 2018).

3. ILS monitored through independent, regular, and calibrated low-pressure gas-cell measurements (Hase et al. (1999); Hase (2012); Fig. 5 of García et al. (2021b)), whereby the independence of FTIR gas retrievals and the instrumental characterisation is ensured.

The stratospheric gases are more sensitive to the ILS treatment and its temporal behaviour than the tropospheric ones, since the full width at half maximum of their sharp absorption lines (absorptions taking place at low pressure) and of ILS have similar magnitudes (Takele Kenea et al., 2013; Sun et al., 2018). Given its key role in atmospheric chemistry, the ILS effects are of special relevance for the analysis of ozone ($O_3$) (e.g. Vigouroux et al., 2008; Cuevas et al., 2013; De Mazière et al., 2018). Recently, Sun et al. (2018) have documented that a typical ILS degradation of 10% may produce changes in the $O_3$ TCs by 2%, but the resulting disturbances of the vertical profile are considerably larger (between ±20%) and altitude-dependent.





These values become especially important when compared to the rather small signals of $O_3$ recovery obtained from long-term observations or projected from chemistry climate models. As summarised in the latest WMO/UNEP (World Meteorological Organization/United Nations Environment Programme) report (WMO, 2018), no significant trend has been detected in global (60ºS–60ºN) $O_3$ TCs over the 1997–2016 period and, outside the polar regions, only upper stratospheric $O_3$ has been found to increase significantly by 0.1–0.3% per year since 2000. Consistent estimates are predicted by climate models (e.g. Hegglin and

Shepherd, 2009; Li et al., 2009; Steinbrecht et al., 2017), pointing to a vertical stratification of the $O_3$ recovery in response to the combined effects of an acceleration of Brewer-Dobson circulation, the stratospheric cooling induced by increasing greenhouse gases concentrations, and the leveling off of anthropogenic $O_3$ depleting substances.

Within the FTIR community considerable efforts have been paid on the development, optimisation, and validation of retrieval strategies for atmospheric $O_3$ monitoring (e.g. Barret et al., 2002; Schneider et al., 2005, 2008a; Schneider et al., 2008b;

Schneider and Hase, 2008; Vigouroux et al., 2008; Lindenmaier et al., 2010; García et al., 2012, 2014; Vigouroux et al., 2015; Zhou et al., 2020; García et al., 2021a). These works were mainly focused on investigating the optimal selection of the $O_3$ spectral absorption lines or the inversion settings used, such as the a-priori information, the different constrains, or the inclusion of an additional atmospheric temperature profile retrieval. Nonetheless, the influence of instrumental characterisation has not been addressed in detail yet. This is the focus of this paper, where the impact of the different approaches for characterising the

ILS (aforementioned as approaches 1-3) are examined taking FTIR $O_3$ products as an example. The study has been performed at the subtropical Izaña Observatory (IZO), where since 1999 ground-based FTIR observations have been carried out coincidentally with other independent $O_3$ measurement techniques. In addition, the ILS function of the IZO FTIR spectrometers has been routinely monitored by means of independent gas-cell measurements since 1999. These two facts make IZO a unique place for developing and documenting the reliability of new $O_3$ retrieval strategies from ground-based FTIR spectrometry. In

this context, this paper is structured as follows: Section 2 describes the Izaña Observatory and its $O_3$ programme, while Section 3 presents the IZO FTIR observations, describing the monitoring of the ILS function, the $O_3$ and ILS retrieval strategies used in this work, and their theoretical characterisation in terms of vertical sensitivity and expected uncertainties. Section 4 addresses the comparison of the ILS time series determined from gas-cell measurements and those simultaneously retrieved with the $O_3$ concentrations from the measured solar absorption spectra. Section 5 assesses the impact of the different ILS treatments on the

FTIR $O_3$ products by comparing to independent datasets (Brewer TC observations and Electrochemical Concentration Cell, ECC, sondes). Finally, Section 6 summarises the main results and conclusions drawn from this work.

## 2 Izaña Observatory and its Ozone Programme

The Izaña Observatory (IZO) is a subtropical high-mountain station, managed by the Izaña Atmospheric Research Center (IARC, https://izana.aemet.es), belonging to the State Meteorological Agency of Spain (AEMet, www.aemet.es). It is on the

island of Tenerife in the North Atlantic Ocean (28.3ºN, 16.5ºW) and located on the top plateau of Izaña mountain at 2373 m a.s.l.. From a climatic point of view, IZO is located below the descending branch of the Northern subtropical Hadley cell, under a quasi-permanent subsidence regime, and typically above a well-established thermal inversion layer. Moreover, the


cities and the moderate industrial activity of the island are concentrated on the coast, thereby the observatory is not affected by significant local and regional pollution contributions (especially during night-time, when the subsidence regime prevails). The

combination of these factors ensures clean air and clear-sky situations during most of the year and offers excellent conditions for atmospheric composition monitoring. As a result, since many years IZO has been engaged in several international atmospheric and environmental activities, and research networks. Refer to Cuevas et al. (2019) for more details about IZO and its atmospheric monitoring programmes.

## 2.1 FTIR Programme

Within the IZO's research activities, the FTIR programme was established in 1999 in the framework of a collaboration between the AEMet-IARC and the KIT (Karlsruhe Institute of Technology), with the main goals of the long-term monitoring of atmospheric gas composition and the validation of space-based observations and climate models. Since then two Bruker high-resolution FTIR systems have been operated at IZO contributing to NDACC: an IFS 120M from 1999 to 2005 and an IFS 120/5HR from 2005 until the present moment.

The FTIR $O_3$ measurements within NDACC are retrieved from the measured solar absorption spectra in the 990-1015 $cm^{-1}$ spectral region by using a potassium bromide (KBr) beam splitter, and a cooled mercury cadmium telluride (MCT) detector. To resolve the narrow $O_3$ absorption lines, the solar spectra have been acquired at the high spectral resolution of 0.0036 $cm^{-1}$ (250 cm of maximum optical path difference, OPD, $OPD_{max}$) until April 2000, and of 0.005 $cm^{-1}$ ($OPD_{max}$=180 cm) onward. The IFS 120M's field-of-view (FOV) angle has varied between 0.17° and 0.29° depending on the measurement period, while

for the IFS 120/5HR it has been always limited to 0.2°, which are considerably lower than the solar diameter of 0.5°. For this study the 20-year $O_3$ measurements taken from 1999 to 2018 have been used. Refer to García et al. (2021b) for a detailed description of the FTIR spectrometry activities at IZO.

## 2.2 Brewer and ECC Sonde Programmes

At IZO $O_3$ TC observations have been also continuously taken by Brewer spectrometers since 1991. In 2001 the IZO Brewer

activities were accepted by NDACC and, two years later, the RBCC-E (Regional Brewer Calibration Centre Europe, www.rbcc-e.org) of the WMO/GAW (Global Atmospheric Watch) programme was established at the observatory. Although the full uncertainty budget is on development, provisional results indicate that the IZO RBCC-E reference instruments can provide $O_3$ TCs with a total uncertainty (standard uncertainty, k=1) between 1.2-1.5% (Gröbner et al., 2017). For the current study the $O_3$ TCs of the permanent instrument Brewer#157 have been used, which were computed following the data processing of the

EUBREWNET network (León-Luis et al., 2018; Redondas et al., 2018).

The latest extension of the IZO $O_3$ programme is the uptake of regular $O_3$ sonde observations, which were initiated in November 1992 and included within NDACC in 2001. Since then, an $O_3$ sounding has been performed once a week from the Santa Cruz Station (30 km north-east of IZO, 36 m a.s.l.) until 2011, when they were moved to a launch site at the Botanic Observatory (13 km north of IZO, 114 m a.s.l.). The $O_3$ sounding is based on ECC that senses $O_3$ as it reacts with a dilute

solution of potassium iodide (KI) to produce an electrical current proportional to the atmospheric $O_3$ concentration (Komhyr,





1986). Until September 1997 the $O_3$ sonde model SPC-5A was used, then it was updated to the SPC-6A model (SPC, 1996) with a cathode sensing solution type SST1.0 (1.0% KI and full pH-buffer). The sounding provides $O_3$ (mPa) profiles, from the ground to the burst level (generally between 30 and 35 km), with a resolution of 0.01 mPa and accuracy of $\pm5$–15% in the troposphere and $\pm5$% in the stratosphere (WMO, 2014).

Note that for the purpose of this paper both Brewer and ECC sonde databases fully cover the entire FTIR 1999-2018 period.

Although they are not used in this work, the IZO $O_3$ programme also includes DOAS (Differential Optical Absorption Spectroscopy) observations, performed within NDACC since 1999, and ground-level $O_3$ records, taken in the framework of the WMO/GAW programme since 1987. More details about these measurement techniques are given in Gil-Ojeda et al. (2012) and Cuevas et al. (2013, 2019).

## 3    FTIR ILS and Ozone Observations

### 3.1    Monitoring of the ILS Function

At IZO much effort has been paid on the instrumental characterisation of the two FTIR spectrometers. Since 1999 the ILS function has been routinely monitored about every two months using low-pressure $N_2O$-cell measurements and the software LINEFIT (v14.5). The LINEFIT package allows the deviation of the measured ILS from the ideal one to be determined (Hase

et al., 1999). It retrieves a complex modulation efficiency (ME) as a function of the OPD, whose real part represents the width of the ILS (ME amplitude, MEA), while its imaginary part accounts for the degree of asymmetry of the ILS (ME phase error, PE).

The ILS parameters used in the FTIR $O_3$ retrievals at IZO are estimated from the $N_2O$-cell measurements using two broad micro-windows, combining saturated and un-saturated $N_2O$ absorbing lines between 1235.0-1279.5 and 1291.8-1301.9 cm$^{-1}$,

and following the approach suggested by Hase (2012). In the Hase's approach the MEA at zero OPD (ZPD) is kept fixed to unity, while the PE is not constrained at ZPD. Here the Hase's method has been modified by considering a clamped PE retrieval at ZPD (i.e. the PE is kept fixed to zero at ZPD). This modification has proved to allow for a superior interpretation of the measured $O_3$ absorption lines and, thus, to offer a better reconstruction of ILS than the retrievals with free PE at ZPD. The ILS time series evaluated assuming a non-clamped PE retrieval (i.e. free PE) at ZPD, and the comparison of the theoretical

performance of the $O_3$ retrievals between the clamped and non-clamped PE retrievals, are included and discussed in Appendix A.

Figure 1((a)-(f)) displays two exemplary contrasting ILS retrievals showing, on the one hand, the noticeable ILS degradation of the IFS 120M spectrometer in 2003 and, on the other hand, the proper alignment of the IFS 120/5HR after 2008. For both instruments the overall response of the ILS estimates on the actual instrumental performance is rather consistent. As observed

in the rows of the averaging kernel matrix **A** obtained in the ILS inversion procedure, the sensitivity of the MEA retrieval slightly increases towards maximum OPDs, which ensures enough sensitivity to resolve the narrower $O_3$ absorption lines, while the PE sensitivity is larger towards ZPD to properly characterise the instrumental asymmetry issues. Nonetheless, the comparison between the two instruments also confirms the improved sensitivity of the IFS 120/5HR as compared to the IFS



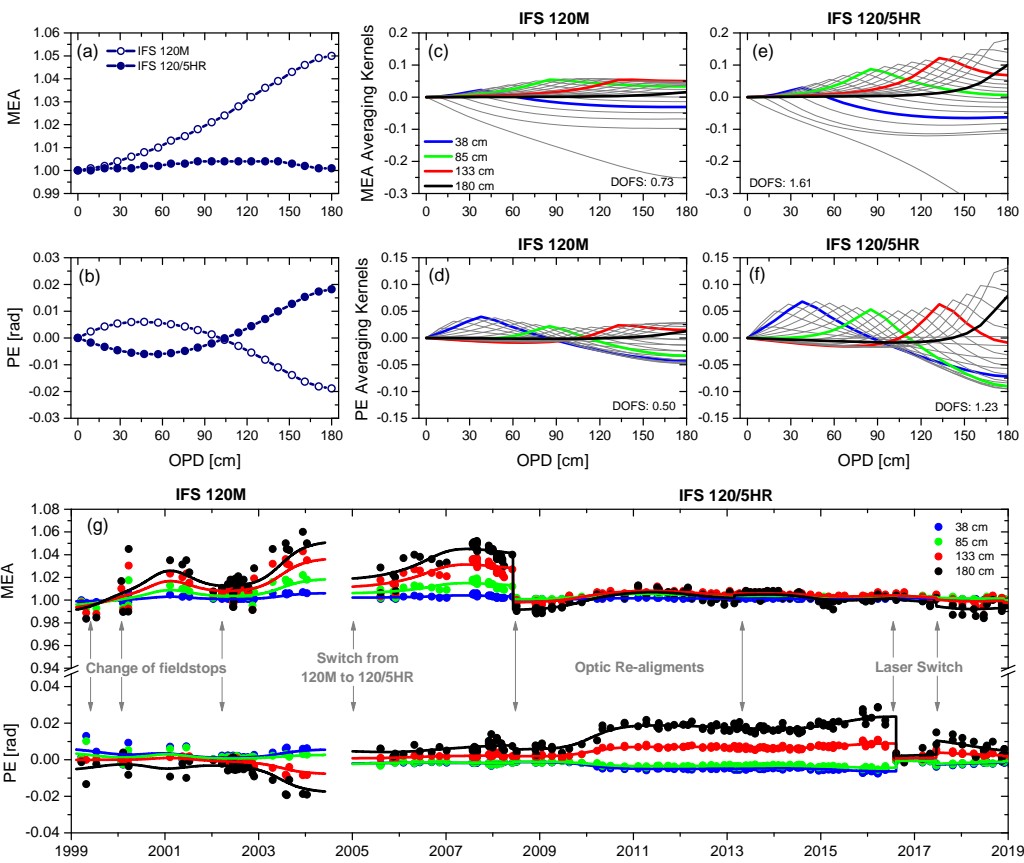

**Figure 1.** Example of the (a) normalised MEA and (b) PE (rad) retrievals as a function of the OPD for the IFS 120M (N$_2$O-cell measurement taken on $3^{rd}$ July 2003) and for the IFS 120/5HR (N$_2$O-cell measurement performed on $20^{th}$ March 2013). (c) and (d) averaging kernel rows of the MEA and PE retrievals for the cell example of the IFS 120M. (e) and (f) same as (c) and (d), but for the IFS 120/5HR. Shown are also the degrees of freedom for signal, DOFS. (g) Time series of the normalized MEA and PE values at four OPD (38, 85, 133 and 180 cm) between 1999 and 2018. Data points represent individual N$_2$O-cell measurements and solid lines depict the smoothed MEA and PE curves. The grey solid arrows indicate punctual interventions on the IZO FTIR instruments: changes of fieldstops between 1999 and 2004, switch from the IFS 120M to the IFS 120/5HR in January 2005, optic re-alignments in June 2008 and February 2013, and internal laser replacements in August 2016 and June 2017.





120M due to a poorer signal-to-noise ratio in the cell spectra of the latter. As a result, the number of independent pieces of ILS
information that can be estimated (given by the trace of **A** matrix, the so-called "degrees of freedom for signal", DOFS) for
the IFS 120/5HR are almost greater by a factor of two than those obtained for the IFS 120M instrument. For the cell examples
of Figure 1, the MEA DOFS is 0.73 and 1.61 for the IFS 120M and the IFS 120/HR, respectively, while the PE DOFS is 0.50
and 1.23, respectively.

The continuous monitoring of the ILS function is critical to control the instrumental alignment, its temporal stability as well
as to detect instrumental issues. The ILS time series at IZO (Figure 1 (g)) shows a few jumps due to punctual interventions
on the FTIR instruments between 1999 and 2018: (1) different changes of fieldstops between 1999 and 2004; (2) the switch
from the IFS 120M to the IFS 120/5HR instrument in January 2005; (3) two optic re-alignments of the IFS 120/5HR system
in June 2008 and February 2013; and (4) two replacements of the spectrometer's internal laser in July 2016 and July 2017.
Figure 1 also illustrates that, besides suffering from a greater level of spectral noise in the cell and atmospheric measurements,
the ILS of the IFS 120M spectrometer is less stable over time than the ILS of the IFS 120/5HR. After the first re-alignment
in 2008, the IFS 120/5 HR's MEA shows a deviation of an ideal instrument of only ∼1% throughout the whole OPD range,
that remains until present, while for the IFS 120M deviations of up to 5% are observed. Regarding the PE time series, subtle
asymmetries (±0.02 rad) are observed from 2003 onward in the IFS 120M and between 2010 and 2016 in the IFS 120/5HR.
The former was due to the IFS 120M instrumental degradation, while the latter was likely introduced in the replacement of the
interferometer's scanner motor at the end of 2009, and was properly corrected with the internal laser switch in July 2016. As
discussed by Hase (2012), finite divergence and misalignment of the internal reference laser might slightly distort the sampling
positions and, thus, impacting PE.

From the analysis of the ILS evolution three periods with different features affecting the IZO FTIR measurements clearly
emerge: (1) 1999-2004, in spite of $N_2O$-cell measurements were routinely carried out, the ILS estimation is imprecise due to
the instability of the IFS 120M spectrometer; (2) 2005-May 2008, although the IFS 120/5HR instrument exhibits a gradual
temporal drift, the ILS function is well-assessed; and (3) June 2008-2018, the IFS 120/5HR instrument is well-characterised
and optically well-aligned (ILS is nearly nominal). Thereby, these three periods will be independently analysed in the present
work in order to examine the impact of the instrument's status on the FTIR $O_3$ products.

### 3.2 Ozone and ILS Retrieval Strategies

In order to determine how the ILS characterisation affects the FTIR $O_3$ products, the retrieval strategy that theoretically and
experimentally showed the best performance in García et al. (2021a) has been modified to assess different ILS treatments
(the so-called 5MWs set-up). In this strategy the $O_3$ volume mixing ratio (VMR) profiles are estimated from the measured $O_3$
absorption lines in five single micro-windows between 991 and 1014 $cm^{-1}$ by means of the inversion code PROFFIT (PROFile
FIT, Hase et al., 2004). The $O_3$ retrieval is performed using an ad-hoc Tikhonov-Philips slope constraint (TP1) on a logarithmic
scale. The a-priori VMR profiles for $O_3$ and all interfering species considered ($H_2O$, $CO_2$, $C_2H_4$, $^{686}O_3$, $^{668}O_3$, $^{676}O_3$ and
$^{667}O_3$) correspond to the climatological simulations from WACCM-version 6 (Whole Atmosphere Community Climate Model,



Marsh et al., 2013), while the spectroscopic parameters are taken from the HITRAN 2008 database with a 2009 update for $H_2O$ (www.cfa.harvard.edu). Refer to García et al. (2021a) for more details about the $O_3$ retrieval strategy.

The strategies considered to deal with the ILS function are listed in Table 1, which are based on different approaches traditionally used by the FTIR community (e.g. Schneider and Hase, 2008; Vigouroux et al., 2008; Vigouroux et al., 2015; Sun et al., 2018). Set-up 5A considers the ILS time series obtained from independent $N_2O$-cell measurements, and evaluated with LINEFIT-v14.5 (Figure 1). The set-up 5B assumes an ideal ILS function, i.e., both the MEA and the PE are set to 1 and 0, respectively, for the whole OPD range. The configuration 5C only retrieves the PE parameter, which is fitted to a constant value throughout the whole OPD range, while the MEA is considered to be ideal (i.e. equal to 1). The set-ups 5D and 5F only estimate the MEA parameter (also so-called effective apodization parameter, EAP, fit), which is calculated by using a second-order polynomial fit of OPD as follows:

$$MEA = 1 + (\alpha - 1)(x/OPD_{max}) + \beta(x/OPD_{max})^2 \tag{1}$$

where $\alpha$ and $\beta$ are the linear and square parameters, respectively, $OPD_{max}$ is equal to 180 cm, and the MEA is sampled at 20 equidistant positions in the interval $\mathbf{x} = 0, ..., OPD_{max}$. The 5D set-up assumes that the MEA linearly varies with the OPD, whereby only the linear parameter $\alpha$ is retrieved simultaneously with the $O_3$ concentrations, while the 5F strategy considers a second-order polynomial dependency and, thus, both the linear and square parameters are estimated. The configurations 5E and 5G optimise the set-ups 5D and 5F, respectively, by including the PE fit, which is retrieved similarly to the set-up 5C (i.e. a constant value throughout the whole OPD range). All strategies 5C, 5D, 5E, 5F, and 5G assume an ideal ILS as a-priori information. Finally, in order to account for uncertainties in the assessment of the independent $N_2O$-cell measurements, the set-up 5H uses the cell-derived MEA values, but a simultaneous fit of a PE offset is superimposed to the retrieved path-dependent cell-derived PE.

As reported by previous works (Schneider and Hase, 2008; Schneider et al., 2008a; García et al., 2012; García et al., 2021a), the quality of the FTIR $O_3$ products can be significantly improved by including a simultaneous atmospheric temperature profile retrieval. However, this enhancement can be only achieved provided the FTIR spectrometer is well-characterised and stable over time. For unstable instruments, the temperature fit has been found to exhibit a strong negative impact on the $O_3$ retrievals by increasing the cross-interference between the instrumental performance and the temperature retrieval (Schneider and Hase, 2008; García et al., 2021a). In order to examine the combined effect of a simultaneous ILS and temperature retrieval for different instrument's status, an optimal estimation of the atmospheric temperature profile has been also considered in this study. To this purpose, four isolated $CO_2$ absorption lines between 962.80 and 969.60 cm$^{-1}$ have been added to the $O_3$ micro-windows, and the temperature a-priori information and inversion settings have been defined according to (García et al., 2021a, and references therein). The retrieval strategies, whose nomenclatures end with the character "T" include a simultaneous atmospheric temperature profile retrieval (see Table 1).

The FTIR spectra are only recorded when the line of sight (LOS) between the instrument and the Sun is cloud-free. However, to avoid possible contamination of thin clouds or unstable measured spectra, once the $O_3$ retrievals have been computed they are filtered according to the number of iterations at which the convergence is reached, and the fitting residuals between the



**Table 1.** Description of the FTIR $O_3$ retrieval strategies with different ILS characterisations and with and without a simultaneous atmospheric temperature profile retrieval (in brackets the set-ups fitting the temperature profile). The 5A/5AT and 5B/5BT configurations consider the cell-derived and the ideal ILS, respectively. The remaining set-ups fit some of the ILS parameters in the $O_3$ retrieval procedure. Note that the level of refinement of the ILS retrieval increases from the set-up C to the set-up H.

| Retrieval Strategy | ILS Approach | Linear Parameter ($\alpha$) | Square Parameter ($\beta$) | Phase Error (PE) | Temperature Fit | A-priori ILS |
|---|---|---|---|---|---|---|
| 5A(T) | $N_2O$-cell Retrieval | No | No | No | No(Yes) | Ideal |
| 5B(T) | Ideal | No | No | No | No(Yes) | - |
| 5C(T) | $O_3$ Retrieval | No | No | Yes | No(Yes) | Ideal |
| 5D(T) | $O_3$ Retrieval | Yes | No | No | No(Yes) | Ideal |
| 5E(T) | $O_3$ Retrieval | Yes | No | Yes | No(Yes) | Ideal |
| 5F(T) | $O_3$ Retrieval | Yes | Yes | No | No(Yes) | Ideal |
| 5G(T) | $O_3$ Retrieval | Yes | Yes | Yes | No(Yes) | Ideal |
| 5H(T) | $N_2O$-cell + $O_3$ Retrieval | No | No | Yes | No(Yes) | $N_2O$-cell |

simulated and measured spectra. Then, all $O_3$ datasets are temporally paired to ensure a fair comparison. The coincident and quality-filtered FTIR $O_3$ retrievals amount to 4924 in the 1999-2018 period (~90% of the measured dataset).

### 3.3 Theoretical Quality Assessment

This section presents the theoretical characterisation of the different ILS retrieval strategies based on the evaluation of their performance and expected uncertainties. Table 2 provides an overview of the vertical sensitivity of the different retrievals on real $O_3$ variations (i.e. total $O_3$ DOFS) and of the interpretation of the measured spectra (i.e. fitting residuals).

For all ILS set-ups an averaged total DOFS of ~4 has been consistently obtained, whereby four atmospheric $O_3$ altitude regions can be distinguished by both FTIR instruments (i.e. troposphere, upper troposphere/lower stratosphere - UTLS -, middle, and upper stratosphere). Although the differences between the periods and retrieval strategies lie within the overall variance, the set-ups considering the cell-derived ILS (5A/5AT) seem to present the best performance when the FTIR system is very stable over time (i.e. 2008-2018 period). However the opposite behaviour is observed for the more unstable periods, which is likely due to uncertainties in the cell-derived PE estimates, as documented by the enhancement of sensitivity and the reduction of fitting residuals when the PE is simultaneously fitted in the $O_3$ retrieval procedure (5H/5HT set-ups). For the remaining set-ups the total vertical sensitivity slightly decreases as the ILS set-ups become more sophisticated (from 5C/5CT to 5G/5GT), because the information contained in the measured spectra is then split into the ILS and $O_3$ retrievals (the retrieved state vector space is not perfectly orthogonal). However, in return, the measured spectra are better reproduced. Similar pattern is observed when the temperature profile fit is also included in the retrieval strategy (lower total DOFS and lower fitting residuals are obtained as compared to those set-ups without retrieving the temperature).





**Table 2.** Summary of statistics of the DOFS and fitting residuals for the set-ups 5A/5AT, 5B/5BT, 5C/5CT, 5D/5DT, 5E/5ET, 5F/5FT, 5G/5GT, and 5H/5HT for the periods 1999-2004, 2005-May 2008, and June 2008-2018, and for the entire time series (1999-2018). The fitting residuals are computed as the noise-to-signal ratio for a common spectral region contained in all set-ups (1001.47-1003.04 cm$^{-1}$). Shown are the median (M) and standard deviation ($\sigma$) for each period. The number of quality-filtered measurements is 466, 683, and 3775 for the three periods, respectively, and 4924 for the whole dataset. The strategies showing the best performance (largest DOFS and smallest residuals) are highlighted in bold for each period.

| | DOFS | | | | Residuals (x10$^{-3}$) | | | |
| | 1999-2004 | 2005-2008 | 2008-2018 | 1999-2018 | 1999-2004 | 2005-2008 | 2008-2018 | 1999-2018 |
| Set-up | M, $\sigma$ | M, $\sigma$ | M, $\sigma$ | M, $\sigma$ | M, $\sigma$ | M, $\sigma$ | M, $\sigma$ | M, $\sigma$ |
|---|---|---|---|---|---|---|---|---|
| 5A | 4.29, 0.29 | 4.56, 0.15 | **4.52**, 0.13 | 4.51, 0.18 | 3.51, 1.96 | 2.57, 0.86 | **2.70**, 0.56 | 2.72, 0.93 |
| 5B | 4.25, 0.28 | 4.55, 0.14 | 4.49, 0.12 | 4.49, 0.17 | 3.68, 1.94 | 2.62, 0.84 | 2.78, 0.55 | 2.79, 0.93 |
| 5C | 4.33, 0.30 | 4.57, 0.14 | 4.50, 0.12 | 4.50, 0.17 | 3.49, 1.91 | 2.61, 0.83 | 2.76, 0.54 | 2.77, 0.89 |
| 5D | 4.29, 0.29 | 4.57, 0.15 | 4.49, 0.12 | 4.49, 0.17 | 3.44, 1.97 | 2.55, 0.85 | 2.76, 0.54 | 2.77, 0.92 |
| 5E | **4.35**, 0.31 | **4.58**, 0.15 | 4.49, 0.12 | 4.50, 0.17 | **3.27**, 1.93 | **2.54**, 0.84 | 2.75, 0.54 | 2.75, 0.88 |
| 5F | 4.20, 0.31 | 4.49, 0.16 | 4.41, 0.13 | 4.41, 0.18 | 3.43, 1.97 | 2.55, 0.85 | 2.75, 0.54 | 2.76, 0.92 |
| 5G | 4.27, 0.32 | 4.50, 0.16 | 4.42, 0.13 | 4.42, 0.18 | **3.27**, 1.93 | **2.54**, 0.84 | 2.73, 0.53 | 2.73, 0.88 |
| 5H | **4.35**, 0.30 | **4.58**, 0.15 | **4.52**, 0.13 | **4.52**, 0.17 | 3.37, 1.93 | **2.54**, 0.85 | **2.70**, 0.55 | **2.71**, 0.90 |
| 5AT | 4.09, 0.34 | 4.42, 0.20 | 4.35, 0.15 | 4.35, 0.21 | 3.44, 1.96 | 2.55, 0.86 | 2.68, 0.56 | 2.70, 0.93 |
| 5BT | 4.08, 0.34 | 4.43, 0.20 | 4.33, 0.15 | 4.33, 0.21 | 3.56, 1.95 | 2.59, 0.84 | 2.75, 0.54 | 2.77, 0.92 |
| 5CT | 4.13, 0.35 | **4.45**, 0.20 | 4.34, 0.15 | 4.34, 0.21 | 3.38, 1.92 | 2.57, 0.82 | 2.74, 0.53 | 2.74, 0.88 |
| 5DT | 4.06, 0.36 | 4.42, 0.20 | 4.31, 0.15 | 4.31, 0.22 | 3.42, 1.97 | 2.53, 0.85 | 2.75, 0.54 | 2.76, 0.92 |
| 5ET | 4.12, 0.37 | 4.43, 0.20 | 4.32, 0.15 | 4.32, 0.21 | **3.25**, 1.93 | 2.52, 0.84 | 2.73, 0.53 | 2.73, 0.88 |
| 5FT | 3.95, 0.36 | 4.32, 0.21 | 4.22, 0.16 | 4.22, 0.22 | 3.42, 1.97 | 2.53, 0.85 | 2.74, 0.53 | 2.75, 0.92 |
| 5GT | 4.01, 0.38 | 4.34, 0.21 | 4.23, 0.16 | 4.23, 0.22 | **3.25**, 1.93 | 2.52, 0.83 | 2.72, 0.53 | 2.72, 0.88 |
| 5HT | **4.14**, 0.36 | 4.44, 0.20 | **4.36**, 0.15 | **4.36**, 0.21 | 3.28, 1.93 | **2.51**, 0.85 | **2.67**, 0.55 | **2.68**, 0.90 |

The different ILS strategies have been also evaluated by performing an uncertainty analysis, which follows the formalism given in Rodgers (2000), the NDACC IRWG recommendations (IRWG, 2014), and has been analytically performed by the PROFFIT package. The error budget includes the impact of the spectral measurement noise and errors due to uncertainties in the input parameters accounting for instrumental and model features, which are split into statistical (ST) and systematic (ST) contributions. Particularly, the ILS function estimates are assumed to have an uncertainty of 1% and 0.01 rad for the MEA and PE, respectively. The baseline parameters, accounting for channeling effects and the intensity offsets, are expected to show an error of 0.1%, while the solar pointing issues are limited to 0.001 rad. Regarding atmospheric temperatures, an uncertainty of 2 K up to 50 km a.s.l. and 1 K for higher altitudes has been considered. Possible uncertainties in the determination of solar lines have been also included (an error of 1% and 10$^{-6}$ for intensity and spectral position, respectively). Finally, the O$_3$ spectroscopy





data (intensity and pressure broadening parameters) are assumed to be uncertain by 3%. All these error sources and values can be considered as typical ones on FTIR measurements (e.g. García et al., 2016; Gordon et al., 2022). Refer to García et al.

(2021a, and references therein) for further details on the error estimation.

The propagation of these uncertainty sources for a typical measurement day of the IFS 120/5HR instrument, considering the cell-derived (5A/5AT), ideal (5B/5BT), and the most refined ILS characterisation (5G/5GT), is displayed in Figure 2. The error estimates reveal consistent results for all ILS set-ups. On the one hand, both the statistical and systematic uncertainties do dependent on the $O_3$ spectroscopic signatures due to the increase of the pointing errors and measurement noise at larger

$O_3$ slant column (SC) amounts. On the other hand, the inclusion of a simultaneous temperature retrieval significantly improves the theoretical performance for all FTIR $O_3$ products, as aforementioned. The total statistical errors of $O_3$ TCs are reduced to one third when applying the temperature fit (from ∼1.5-3.0% to ∼0.5-1.0% for $O_3$ SCs between 250 and 3000 DU), while the total systematic contributions drop by 0.1%-0.4% at low and high $O_3$ SCs, respectively. But, the uncertainty analysis also illustrates the cross-interference between the simultaneous ILS and temperature retrievals, which becomes more evident in the

altitude-resolved error patterns (Figure 2 (c) and (d)). For set-ups without the temperature fit, the ILS retrieval (5G) increases the statistical errors of the $O_3$ TCs by significantly worsening the error profiles beyond the middle stratosphere. However, when the temperature and ILS are jointly retrieved with the $O_3$ concentrations (5GT), more precise $O_3$ TCs are expected due to an error reduction in the middle stratosphere layer around 30 km (visible for $O_3$ SCs smaller than 1500 DU in Figure 2 (a)). Although, for simplicity, only the error estimation for the set-ups 5G/5GT is depicted in Figure 2, the remaining strategies

fitting the ILS parameters are found overall to be consistent with these results.

It is worth highlighting that the uncertainty analysis carried out in this work assumes the same uncertainty values for the MEA and PE parameters for all strategies. However, the ILS errors for the set-ups 5B/5BT (ideal MEA and PE) or 5C/5CT (ideal MEA) are expected to be significantly larger than for the 5A/5AT retrievals (cell-derived ILS), especially for the MEA parameter as displayed in Figure 1. As shown by García et al. (2021a), an uncertainty of ∼5% for the MEA parameter may

double the expected total statistical errors provided the temperature fit is taken into account in the $O_3$ retrieval procedure.

## 4   Comparison of the cell-derived and the retrieved ILS

Figure 3 synthesises the comparison between the cell-derived ILS time series for the two IZO FTIR instruments, evaluated from $N_2O$-cell measurements (Figure 1), and those retrieved simultaneously with the $O_3$ concentrations from the most refined ILS set-ups (5G/5GT). The MEA time series as a function of the OPD is shown, as well as the MEA and the PE time series

at an OPD of 133 cm, as an example of the ILS behaviour at large OPDs. Note that the ILS evaluation becomes determinant towards maximum OPDs due to the high spectral resolution required to resolve the narrow $O_3$ absorption lines.

The ILS comparison proves that the general shape of the cell-derived MEA is reproduced well by the ILS retrieval strategies: the estimated MEA values capture the increasing drift for the IFS 120M instrument and for the IFS 120/5HR spectrometer between 2005 and mid-2008, and its stable evolution since then. However, the fitted MEA shows an artificial annual cycle,

which is more noticeable when the temperature is not simultaneously retrieved, and the instrument is properly aligned and





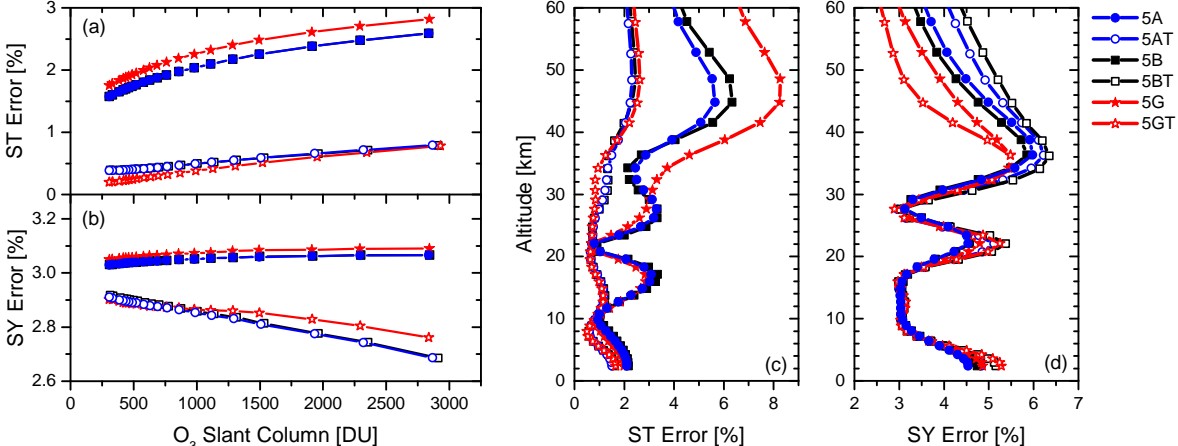

**Figure 2.** (a) Estimated total statistical (ST) and (b) systematic (SY) errors [%] for $O_3$ TCs retrieved from the 5A/5AT, 5B/5BT, and 5G/5GT set-ups as a function of $O_3$ slant column (SC) [DU] for measurements taken on $31^{st}$ August 2007 from solar zenith angles, SZA, between 84° (∼07:00 UT) and 21° (∼13:30 UT). (c) Example of estimated total statistical (ST) and (d) systematic (SY) error profiles [%] for the same set-ups for the spectrum taken on $31^{st}$ August 2007 at a SZA of ∼50°, and an $O_3$ SC of 390 DU. The total errors are computed as the square root of the quadratic sum of all ST and SY error sources considered.

stable (after the first re-alignment in 2008, zoom in Figure 3 (e)). In addition, fitting the ILS parameters produces, together with a noticeable daily variability, unrealistic MEA retrievals (see, for example, the values at the beginning of 2000 and at the end of 2002 in Figure 3 (e) and (f)). Thereby, the ILS fit seems to misinterpret extreme or anomalous $O_3$ events as instrumental deficiencies. These results point out the existence of a significant cross-interference between the ILS and the $O_3$ concentrations

when both are simultaneously retrieved, meaning that part of the $O_3$ variability on a daily and annual basis are damped when fitting the ILS parameters.

Similarly to the MEA retrievals, the estimated PE values are also able to overall capture the evolution of the IFS 120/5HR instrument together with punctual interventions on the spectrometer (e.g. the scanner's motor change at the end of 2009 and the internal laser switches in 2016 and 2017). As expected, the IFS 120M instrument exhibits more variable retrieved PE values

from the atmospheric spectra than those evaluated from the $N_2O$-cell measurements. Figure 3 (d) also includes the retrieved PE values from the set-up 5H, which superimposes an PE offset to the cell-derived PE values. The excellent agreement between the PE retrievals from atmospheric spectra using an ideal and the cell-derived ILS as a priori information becomes evident (Figure 3 (e) and (f)). However, as shown in Appendix A, larger PE corrections are retrieved from the atmospheric spectra when the cell-derived ILS is evaluated assuming a non-clamped PE retrieval (especially for the IFS 120M instrument). This

fact further corroborates that a clamped PE retrieval at ZPD when evaluating the $N_2O$-cell measurements is a superior choice to characterise the instrumental performance of the NDACC FTIR spectrometers, as outlined in Section 3.1.



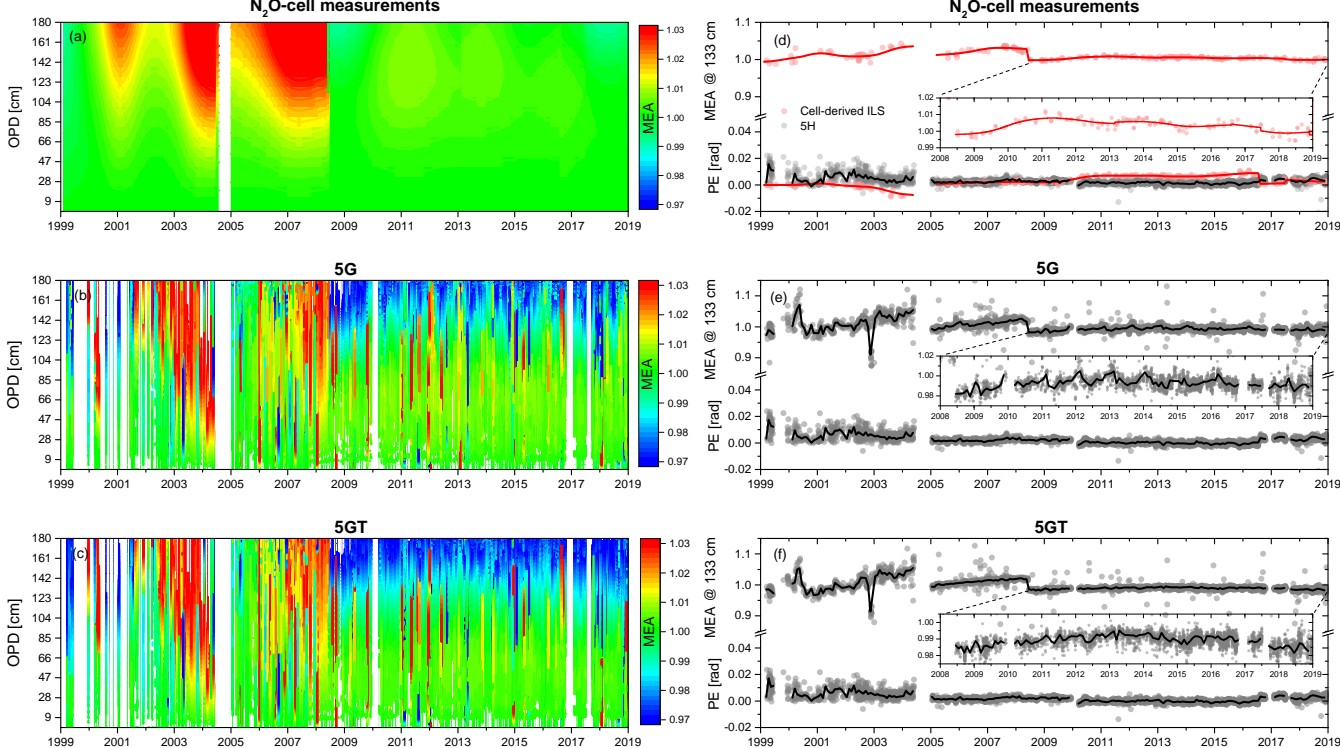

**Figure 3.** Time series of the MEA as a function of the OPD from 1999 to 2018 evaluated from (a) $N_2O$-cell measurements, (b) and (c) $O_3$ retrievals (5G/5GT set-ups). (d), (e), and (f) same as (a), (b), and (c), but for the MEA and PE at an OPD of 133 cm. The PE values retrieved from the 5H set-up are also included in (d). Note that for set-ups 5G/5GT and 5H/5HT the retrieved PE is constant throughout the whole OPD range.

In order to examine in more detail the presence of seasonality in the retrieved ILS time series, Figure 4 displays the averaged annual cycle of the MEA and PE parameters for the more stable IFS 120/5HR period (2009-2018). Similarly to Figure 3, the annual cycles at an OPD of 133 cm for the set-ups 5A/5AT and 5G/5GT are shown. As expected, the cell-derived MEA

and PE values derived from the $N_2O$ cells do exhibit no seasonality. By contrast, a noticeable annual cycle for the retrieved MEA is observed for the 5G/5GT set-ups, although it is partially corrected by the temperature fit. The MEA seasonal range is $8.6 \cdot 10^{-3}$ (0.9% with respect to the mean retrieved MEA) and $4.8 \cdot 10^{-3}$ (0.5% with respect to the mean retrieved MEA) for 5G and 5GT, respectively, which are about 10 and 5 times that of the reference, $0.9 \cdot 10^{-3}$ (0.1% with respect to the mean cell-derived MEA). Regarding the PE parameter, although a subtle seasonal dependence is detected in the retrieved PE values,

no significant differences are observed between the ILS strategies, and no influence of temperature retrieval is found.

Consistent results were obtained for the less sophisticated ILS set-ups: the cell-derived ME and PE values are overall well-reproduced for both FTIR instruments, especially for the IFS 120/5HR periods (data not shown). However, the artificial ILS seasonality for these configurations is less noticeable than for the most refined ILS set-ups (5G/5GT), particularly when the





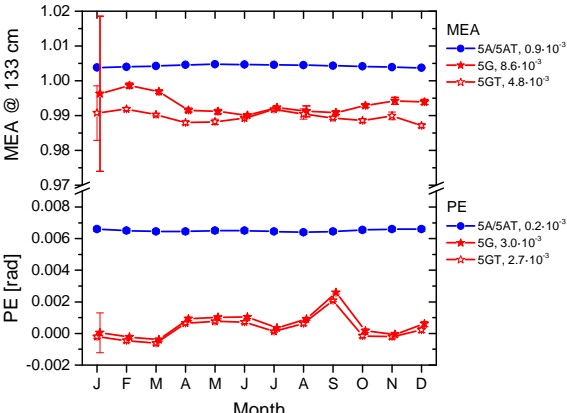

**Figure 4.** Averaged annual cycle of the MEA (at 133 cm) and PE [rad] from 2009 to 2018 for the set-ups 5A/5AT, and 5G/5GT. For 5A/5AT the PE is given at 133 cm, while for the other strategies it corresponds to a unique value throughout the whole OPD range. The annual range (annual maximum minus annual minimum) is shown in the legend for each set-up. The error bars in the seasonal cycles are the standard errors of the mean: $2 \times \sigma/\sqrt{N}$, with $\sigma$ the standard deviation and $N$ the number of monthly measurements.

temperature retrieval is not considered. The MEA seasonal range is limited to 0.4% (5D/5E) and 0.5% (5DT/5ET) with respect to the mean retrieved MEA for the 5E/5ET set-ups, respectively.

In case of gas-cell measurements are not available and the independence of ILS and target gas retrievals is pursued, an alternative approach might be to retrieve the ILS information from atmospheric trace gas retrievals with well-known vertical distribution (Vigouroux et al., 2015). To assess the viability of this strategy, the ILS parameters have been also evaluated from the measured absorbing lines of very stable tropospheric and stratospheric gases, such as $CO_2$ and HF, respectively. Nonetheless, as discussed in Appendix B, both strategies produce unrealistic ILS estimates and a strong artificial annual cycle in the retrieved MEA and PE values, therefore they have been discarded in the subsequent $O_3$ assessment. Consequently, in absence of independent ILS measurements, the most promising approach to characterise the instrumental performance might be to apply a retrieval of the ILS parameters simultaneously to the atmospheric temperature and $O_3$ profiles (provided the FTIR instrument is stable over time).

## 5 Comparison to Reference Observations

### 5.1 FTIR and Brewer Ozone Total Columns

The quality assessment of the different ILS set-ups has been addressed by a straightforward comparison to coincident Brewer observations, acquired within a temporal window of $\pm 5$ minutes around the FTIR measurements. This matching criterion leads to 1958 coincidences in the 1999-2018 period. As observed in the 20-year time series of the relative differences (RD, FTIR-Brewer) for the set-ups 5A/5AT, 5B/5BT, and 5G/5GT (Figure 5), the differences between the ILS set-ups become more





significant when the temperature is simultaneously estimated, corroborating the cross-interference between both retrievals. The most remarkable discrepancies occur for the IFS 120M period and between 2005 and mid-2008 for the IFS 120/5HR instrument, when the behaviour of both systems was far from ideal. In fact, when assuming the ILS to be ideal (5BT set-up) the FTIR O$_3$ TCs seem to absorb the actual ILS temporal degradation as documented by the consistent increasing drift observed

in the RD and cell-derived MEA time series (2003-2008 period in Figure 1 and Figure 5 (b)). By contrast, when the IFS 120/5HR behaves almost as an ideal instrument (2008-2018 period) no significant discrepancies were found between using the cell-derived or an ideal ILS (with and without temperature retrieval). This fact further confirms the reliability of the ILS estimates from independent gas-cell measurements.

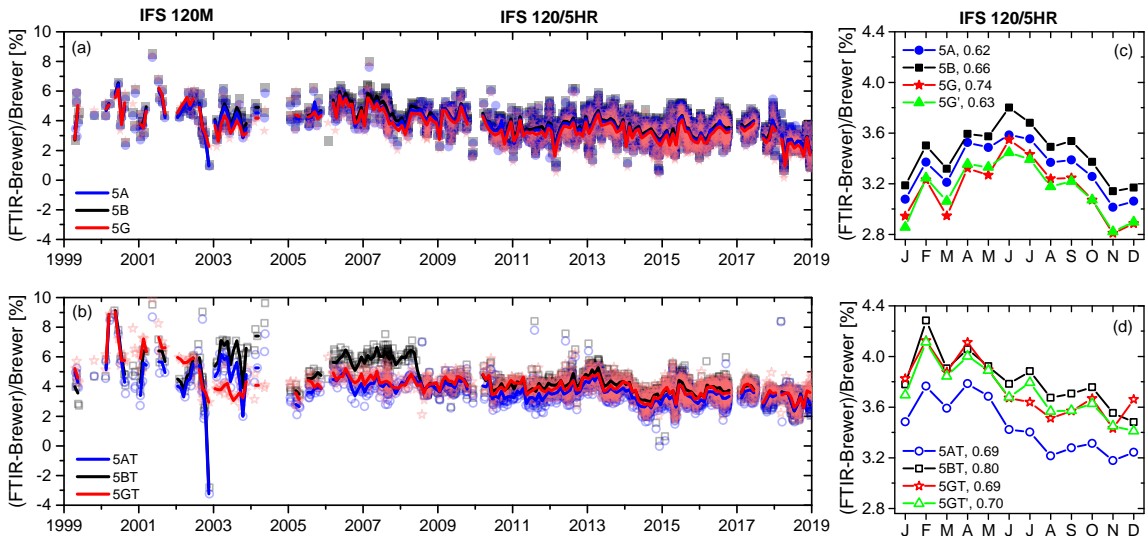

**Figure 5.** Summary of Brewer-FTIR comparison from 1999 to 2018. (a) Time series of the relative differences RD for the set-ups 5A, 5B, and 5G, which are calculated as RD[%]=100×(O$_3$ TC$_X$ − O$_3$ TC$_Y$)/O$_3$ TC$_Y$, where X and Y means FTIR and Brewer, respectively. (b) Same as (a), but for the set-ups 5AT, 5BT, and 5GT. (c) Averaged annual cycles of the RD for the 2009-2018 period for the set-ups 5A, 5B, and 5E. (d) Same as (c), but for the set-ups 5AT, 5BT, and 5GT. The averaged annual cycles for the ILS strategies 5G' and 5GT' have been also included in figures (c) and (d), respectively. The solid lines in (a) and (b) correspond to the monthly median values. The legend in (c) and (d) shows the seasonal amplitude of each cycle.

Figure 5 also illustrates that, although the scatter of RD is significantly improved by the temperature retrieval, these strategies

present more extreme RD values, which are larger for those set-ups without fitting the ILS. Note, for example, the correlation between the unrealistic retrieved MEA values at the beginning of 2000 and at the end of 2002 (Figure 3 (e) and (f)) and the extreme RD values only obtained for those set-ups not retrieving the ILS parameters (especially in Figure 5 (b)). This behaviour is likely due to the ILS retrievals are absorbing the most imprecise FTIR O$_3$ retrievals, as mentioned in Section 4. But, also,





the extreme RD values may indicate measurement days with an unusual temperature vertical stratification, which might be
wrongly captured by the Brewer and FTIR products assuming a fixed temperature (and pressure) profile (García et al., 2021a).

An overview of the FTIR-Brewer comparison for all ILS set-ups, distinguishing between the instruments and periods, is given in Table 3. Consistently for both instruments, when the temperature fit is not considered in the retrieval strategy, the set-ups using the cell-derived MEA values (5A and 5H) seem to offer the best agreement with respect to Brewer data (less dispersion, greater accuracy, and correlation). Nonetheless, the inclusion of the simultaneous temperature estimation enhances
the performance of those strategies fitting the MEA parameter (5DT, 5ET, 5FT, and 5GT set-ups), confirming the results obtained in the theoretical quality assessment (Section 3.3). Note also that the PE retrieval has a minor impact on the $O_3$ products for both FTIR instruments. Considering the IFS 120/5HR periods as reference, the combined strategies provide the most precise $O_3$ TCs with a scatter in the RD of ~0.5-0.6% (5DT, 5ET, 5FT, and 5GT), while it increases up to ~0.6-0.7% when using the cell-derived MEA values (5AT and 5HT), and up to ~0.7-0.8% when the MEA is assumed to be ideal (5BT
and 5CT). However, this improvement is not good enough to completely cancel out the negative cross-interference between the temperature retrieval and instrumental performance for an unstable FTIR spectrometer such as the IFS 120M. In that case, the most precise $O_3$ TCs are obtained considering the cell-derived ILS data and without performing a simultaneous temperature fit in the $O_3$ retrieval procedure.

Regarding systematic differences, the FTIR-Brewer bias ranges from 3% to 6%, which is compatible with errors of ~3%
in the $O_3$ infrared spectroscopy as documented by the theoretical uncertainty analysis. Such differences are consistent with previous studies (e.g. Schneider et al., 2008a; García et al., 2012) and with ultraviolet/infrared/microwave intercomparison experiments carried out in laboratory (e.g. Piquet-Varrault et al., 2005; Gratien et al., 2010; Tyuterev et al., 2019). Nonetheless, the simultaneous ILS and temperature retrievals as well as the instrument status also contribute to the observed offset. As shown in Table 3, the combined (ILS and temperature) approaches increase the differences with respect to Brewer data as compared
to the usage of the cell-derived ILS values. Note that the reduction of the FTIR-Brewer bias between the two IFS 120/5HR periods is likely introduced by punctual interventions on the FTIR spectrometer (García et al., 2014; García et al., 2021a).

The $O_3$ concentrations are expected to be impacted by the different ILS approaches at a seasonal scale. To examine this effect Figure 5 also includes the averaged annual cycle of the RD for the more stable IFS 120/5HR period (2009-2018). Independently of the temperature treatment, the $O_3$ strategies using the cell-derived MEA values show the lowest RD seasonal amplitudes.
They vary from 0.62% (5A) to 0.69% (5AT), and from 0.57% (5H) to 0.62% (5HT), while for the most refined ILS set-up the annual peak-to-peak amplitude ranges from 0.74% (5G) to 0.69% (5GT). The latter is even larger than those observed when assuming an ideal MEA value and the simultaneous temperature retrieval is not considered (0.66% for the set-ups 5B and 5C). These results suggest that the artificial annual cycle of the retrieved ILS parameters comes along with a misinterpretation of the retrieved seasonal cycle of $O_3$, as already pointed out by the comparison of the cell-derived and the retrieved ILS time series
(Section 4): the simultaneous ILS fitting leads to an underestimation of the actual $O_3$ seasonality.

In order to reduce this seasonal artefact an additional approach has been assessed, which is based on the retrieved MEA and PE parameters from the most sophisticated ILS configuration (5G/5GT) (so-called 5G'/5GT' set-ups). Firstly, the estimated ILS parameters are averaged on an annual basis in the three periods analysed independently. Then, the annually-averaged





**Table 3.** Summary of statistics for the Brewer-FTIR comparison for the set-ups 5A/5AT, 5B/5BT, 5C/5CT, 5D/5DT, 5E/5ET, 5F/5FT, 5G/5GT, and 5H/5HT: median (M, in %) and standard deviation ($\sigma$, in %) of the relative differences and Pearson correlation coefficient (R) for the periods 1999-2004, 2005-May 2008 and June 2008-2018, and for the entire time series (1999-2018). The number of coincident FTIR-Brewer measurements is 89, 169, and 1722 for the three periods, respectively, and 1980 for the whole dataset. Note that the set-ups 5G' and 5GT' stand for the FTIR $O_3$ data estimated using the retrieved ILS parameters from the set-ups 5G and 5GT, respectively, but averaged on a annual basis (see details in the text). The strategies showing the best performance (smallest M and $\sigma$, and largest R) are highlighted in bold for each period.

| | 1999-2004 | 2005-2008 | 2008-2018 | 1999-2018 |
|---|---|---|---|---|
| Set-up | M[%], $\sigma$[%], R | M[%], $\sigma$[%],R | M[%], $\sigma$[%], R | M[%], $\sigma$[%], R |
| 5A | 4.33, **1.14, 0.972** | 4.47, **0.81, 0.975** | 3.39, 0.82, 0.982 | 3.49, **0.91, 0.977** |
| 5B | 4.50, **1.14, 0.972** | 4.75, 0.82, 0.974 | 3.51, 0.83, 0.982 | 3.62, 0.93, 0.976 |
| 5C | 4.45, 1.14, 0.971 | 4.75, 0.82, 0.974 | 3.51, 0.83, 0.982 | 3.61, 0.93, 0.976 |
| 5D | 4.22, 1.17, 0.968 | 4.45, 0.89, 0.970 | 3.43, 0.86, 0.980 | 3.51, 0.94, 0.976 |
| 5E | **4.15**, 1.17, 0.968 | 4.45, 0.89, 0.969 | 3.42, 0.86, 0.980 | 3.50, 0.94, 0.976 |
| 5F | 4.17, 1.15, 0.969 | **4.37**, 0.88, 0.971 | 3.23, 0.83, 0.981 | 3.34, 0.94, 0.975 |
| 5G | 4.18, 1.16, 0.969 | 4.38, 0.88, 0.971 | 3.24, 0.83, 0.981 | 3.34, 0.94, 0.975 |
| 5G' | 4.29, 1.19, 0.967 | 4.42, 0.84, 0.973 | **3.22, 0.81, 0.982** | **3.33**, 0.92, 0.976 |
| 5H | 4.25, **1.14, 0.972** | 4.46, **0.81, 0.975** | 3.38, 0.82, 0.982 | 3.48, **0.91, 0.977** |
| 5AT | 4.91, 1.60, 0.953 | **4.13**, 0.61, 0.985 | **3.43**, 0.67, 0.988 | **3.52**, 0.80, 0.982 |
| 5BT | 5.63, 1.72, 0.943 | 5.88, 0.79, 0.974 | 3.81, 0.70, 0.987 | 3.91, 1.00, 0.972 |
| 5CT | 5.65, 1.73, 0.941 | 5.89, 0.80, 0.974 | 3.81, 0.70, 0.987 | 3.91, 1.00, 0.972 |
| 5DT | 5.20, 1.47, 0.947 | 4.34, **0.52, 0.989** | 3.59, 0.64, 0.988 | 3.68, 0.79, 0.982 |
| 5ET | 5.18, **1.47, 0.948** | 4.35, 0.53, 0.989 | 3.59, 0.64, 0.988 | 3.68, 0.79, 0.982 |
| 5FT | 5.17, 1.48, 0.947 | 4.37, **0.52, 0.989** | 3.73, **0.63, 0.989** | 3.82, **0.76, 0.984** |
| 5GT | 5.13, 1.47, 0.947 | 4.38, **0.52, 0.989** | 3.73, **0.63, 0.989** | 3.82, **0.76, 0.984** |
| 5GT' | 4.90, 1.82, 0.926 | 4.41, 0.55, 0.988 | 3.71, 0.66, 0.988 | 3.78, 0.81, 0.982 |
| 5HT | **4.87**, 1.61, 0.953 | 4.14, 0.61, 0.985 | **3.43**, 0.67, 0.988 | **3.52**, 0.80, 0.982 |

ILS values are used to compute the $O_3$ retrievals in a second step. As shown in Table 3, this approach indeed improves the
agreement with respect to Brewer data for the IFS 120/5HR instrument, when the simultaneous temperature fit is not applied
and the seasonality of the retrieved ILS values is therefore more evident. At a seasonal scale, the peak-to-peak amplitude of
RD is also reduced when the ILS parameters are averaged annually, from 0.74% (5G) to 0.63% (5G') (Figure 5). However,
this strategy does not offer similar results whether the temperature fit is also considered in the retrieval procedure. It slightly
increases the dispersion in the straightforward comparison to the Brewer database (Table 3) and the seasonal amplitude of RD





(from 0.69% to 0.70%). Finally, for the IFS 120M spectrometer this approach has found not to improve the comparability to Brewer data due to the high variability of the retrieved ILS parameters and their remarkable temporal degradation.

## 5.2   FTIR and ECC Ozone Vertical Profiles

The ILS characterisation is especially critical in the retrieval of the $O_3$ vertical distribution, as illustrated in Figure 6. It depicts the median and standard deviation of the relative difference between the FTIR $O_3$ profiles, obtained with different ILS set-ups,

and coincident ECC sondes for the three periods considered. As ECC sondes usually burst between 30 and 35 km, only those sondes with information up to 29 km have been paired to the FTIR profiles within a temporal window of $\pm 3$ hours around the sonde launch ($\sim$12 UT). These matching criteria provide 277 coincidences in the 1999-2018 period. In addition, to account for the limited FTIR vertical sensitivity, the ECC sondes were vertically-smoothed using the averaging kernels obtained in the $O_3$ retrieval procedure (García et al., 2021a, and references therein). Note that, although only the 5A/5AT, 5B/5BT, and 5G/5GTs

set-ups are depicted in Figure 6 for simplicity, Table 4 summarises the comparison statistics for all set-ups at the representative altitudes of the 5, 18, and 29 km. These levels correspond to the altitudes at which the maximum sensitivity is reached by the FTIR system up to the middle stratosphere (Schneider and Hase, 2008; García et al., 2012; Vigouroux et al., 2015; García et al., 2021a).

The impact of the instrumental characterisation exhibit a clear vertical stratification, which depends on the FTIR instrument's

status. Thus, below $\sim$25 km the vertical performance of the different ILS characterisation approaches is quite similar for the two FTIR spectrometers, with scatter values of $\sim$6.4-7.5% up to the UTLS region (5 and 18 km in Table 4) for the IFS 120M and for the 2005-2008 period of the IFS 120/5HR instrument. When the latter was better aligned (2008-2018 period), the dispersion between FTIR and smoothed-ECC sondes is slightly reduced by $\sim$5-6%. As expected, the effect of the temperature retrieval becomes noticeable beyond the upper troposphere.

In the middle stratosphere (above $\sim$25 km), Figure 6 and Table 4 illustrates a differentiated behaviour depending on the FTIR instrument, consistently with the findings documented in the FTIR-Brewer comparison. For the IFS 120/5HR system, the best agreement with respect to ECC data is reached when the ILS, including the MEA parameter, and the temperature fits are simultaneously carried out with the $O_3$ vertical retrieval (5DT, 5ET, 5FT, and 5GT set-ups). In fact, a substantial improvement is documented for the more sophisticated ILS retrievals (5FT/5GT) as compared to the strategy using the cell-derived MEA

values (5AT/5HT). The scatter values at 29 km ranges from 4.5% (5AT/5HT) to 2.6% (5FT/5GT), and from 3.8% (5AT/5HT) to 2.9% (5FT/5GT) for the 2005-May 2008 and June 2008-2018 periods, respectively. Nonetheless, for the IFS 120M instrument, the best agreement is observed for the more refined ILS set-ups not including the temperature retrieval (5F/5G), since the cross-interference between the temperature retrieval and the instrumental performance (ILS uncertainties and spectral measurement noise) worsens its comparability to the ECC data. This negative impact is especially critical whether only the temperature fit is

applied (5AT, 5BT, 5CT, and 5HT set-ups). The scatter values increase from $\sim$4% up to $\sim$7-8% for the IFS 120M, while for the well-aligned IFS 120/5HR period the dispersion increases less than 0.2% at 29 km.

In relation to systematic differences, the altitude-dependence of bias is found to consistently decline as the FTIR instrument is getting stability and a better alignment. The IFS 120M is not sensitive to changes between the ILS set-ups and tempera-

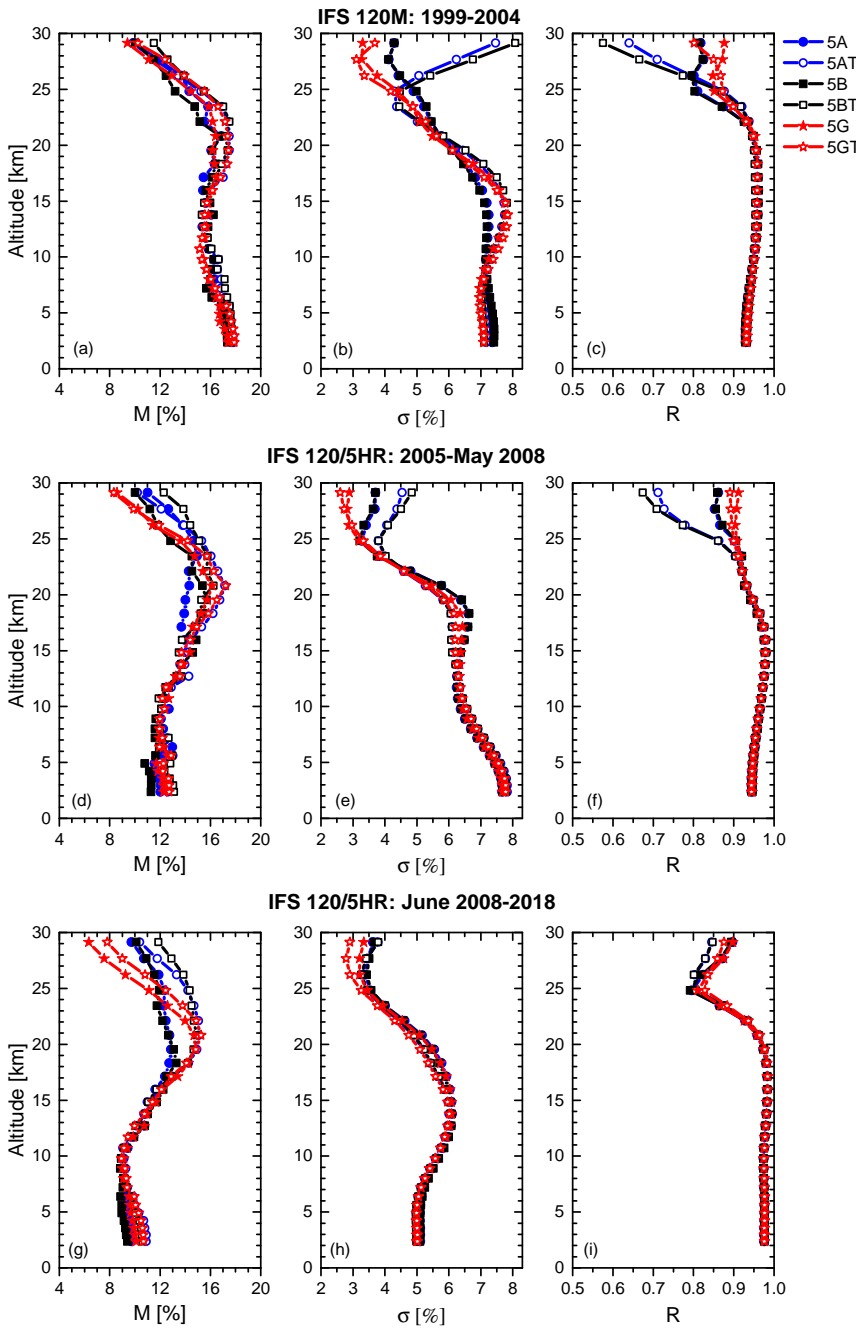

**Figure 6.** Summary of FTIR-smoothed ECC comparison for the periods 1999-2004, 2005-May 2008, and June 2008-2018. (a), (d), and (g) display the vertical profiles of the median (M) RD (FTIR-ECC, in %) for the three periods, respectively. (b), (e), and (h) same as (a), (d), and (g), but for the standard deviation of RD distributions ($\sigma$, in %). (c), (f), and (i) same as (a), (d), and (g), but for the Pearson correlation coefficient. The number of coincident FTIR-ECC measurements is 56, 49, and 167 for the periods 1999-2004, 2005-May 2008, and June 2008-2018, respectively.

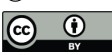


**Table 4.** Same as Table 3, but for the FTIR-smoothed ECC comparison at 5, 18, and 29 km altitude levels. The number of coincident FTIR-ECC measurements is 56, 49, and 167 for the three periods, respectively, and 272 for the whole dataset. The strategies showing the best performance (smallest M and $\sigma$, and largest R) are highlighted in bold for each period.

| Set-up | 1999–2004 M[%], σ[%], R | 2005–2008 M[%], σ[%], R | 2008–2018 M[%], σ[%], R | 1999–2018 M[%], σ[%], R |
|---|---|---|---|---|
| | *FTIR-ECC at 5 km* | | | |
| 5A | 17.12, 7.32, 0.931 | 11.55, 7.44, 0.948 | 9.48, 5.10, 0.976 | 10.91, 6.83, 0.953 |
| 5B | 16.95, 7.37, 0.931 | 10.79, 7.45, 0.949 | **8.98**, 5.11, 0.976 | 10.46, 6.85, 0.953 |
| 5C | 16.78, 7.37, 0.931 | **10.74**, 7.45, 0.949 | 8.99, 5.10, 0.976 | **10.41**, 6.82, 0.930 |
| 5D | 17.17, 7.10, 0.935 | 11.68, 7.49, 0.947 | 9.27, 4.99, 0.977 | 10.68, 6.66, 0.935 |
| 5E | 16.88, 7.08, 0.936 | 11.59, 7.49, 0.947 | 9.28, **4.98, 0.977** | 10.69, 6.63, 0.955 |
| 5F | 17.14, 7.03, 0.936 | 11.81, 7.45, 0.948 | 9.67, 5.00, 0.977 | 11.06, 6.56, 0.956 |
| 5G | 16.73, **7.02, 0.936** | 11.72, 7.45, 0.948 | 9.67, 4.99, 0.977 | 11.01, **6.51, 0.957** |
| 5G* | 16.72, 7.07, 0.936 | 11.49, 7.51, 0.948 | 9.66, 5.08, 0.976 | 11.00, 6.69, 0.955 |
| 5H | 16.96, 7.32, 0.932 | 11.47, 7.44, 0.948 | 9.47, 5.09, 0.976 | 10.79, 6.80, 0.953 |
| | *FTIR-ECC at 18 km* | | | |
| 5A | 16.34, 6.55, 0.959 | 13.93, 6.61, 0.959 | 12.74, 5.78, 0.980 | 13.65, 6.29, 0.973 |
| 5B | 16.43, **6.46**, 0.960 | 15.42, 6.64, 0.958 | 13.29, **5.72, 0.980** | 14.14, 6.27, 0.973 |
| 5C | 16.30, 6.49, **0.961** | 15.37, 6.65, 0.958 | 13.30, **5.72, 0.980** | 14.10, 6.27, 0.973 |
| 5D | 16.59, 6.99, 0.957 | 14.90, 6.46, 0.960 | 13.07, 5.73, 0.980 | 14.03, 6.40, 0.972 |
| 5E | 16.44, 7.06, 0.957 | 14.86, 6.47, 0.960 | 13.07, 5.73, 0.980 | 13.92, 6.41, 0.972 |
| 5F | 16.51, 6.70, 0.958 | 15.24, **6.35, 0.962** | 14.25, 5.74, 0.981 | 14.99, **6.16, 0.974** |
| 5G | 16.31, 6.77, 0.959 | 15.26, 6.36, 0.962 | 14.24, 5.74, 0.981 | 14.88, **6.16, 0.974** |
| 5G* | 16.74, 6.66, 0.960 | 15.39, 6.57, 0.959 | 14.46, 5.80, 0.980 | 14.76, 6.25, 0.973 |
| 5H | **16.18**, 6.58, 0.960 | **13.86**, 6.63, 0.959 | 12.75, 5.79, 0.980 | **13.62**, 6.30, 0.973 |
| | *FTIR-ECC at 29 km* | | | |
| 5A | 9.94, 4.30, 0.818 | 11.02, 3.71, 0.862 | 9.74, 3.63, 0.897 | 9.85, 4.07, 0.861 |
| 5B | 10.04, 4.29, 0.815 | 10.02, 3.71, 0.860 | 10.12, 3.66, 0.894 | 10.07, 4.04, 0.860 |
| 5C | 9.79, 4.35, 0.813 | 10.00, 3.72, 0.859 | 10.11, 3.66, 0.895 | 10.02, 4.06, 0.859 |
| 5D | 10.08, 4.17, 0.826 | 9.46, 3.80, 0.863 | 9.87, 3.62, 0.897 | 9.87, 3.62, 0.897 |
| 5E | 9.82, 4.26, 0.823 | 9.40, 3.81, 0.862 | 9.91, 3.62, 0.897 | **6.82**, 4.01, 0.859 |
| 5F | 9.41, **3.31, 0.876** | 8.59, **2.88, 0.912** | 6.35, 3.34, 0.898 | 7.02, 3.51, 0.880 |
| 5G | 9.21, 3.31, 0.870 | 8.34, 2.89, 0.911 | 6.30, **3.31, 0.899** | 7.00, **3.46, 0.881** |
| 5G* | 9.81, 3.72, 0.863 | 8.24, 3.69, 0.866 | **6.26**, 3.64, 0.892 | 7.10, 4.11, 0.857 |
| 5H | 9.63, 4.36, 0.816 | 11.01, 3.72, 0.861 | 9.66, 3.63, 0.897 | 9.74, 4.09, 0.860 |

| Set-up | 1999–2004 M[%], σ[%], R | 2005–2008 M[%], σ[%], R | 2008–2018 M[%], σ[%], R | 1999–2018 M[%], σ[%], R |
|---|---|---|---|---|
| | *FTIR-ECC at 5 km* | | | |
| 5AT | 17.55, 7.13, 0.935 | 12.93, 7.62, 0.946 | 10.32, 4.99, 0.976 | 11.40, 6.60, 0.955 |
| 5BT | 17.60, 7.06, 0.936 | 12.81, 7.59, 0.946 | 10.05, 4.97, 0.977 | 11.44, 6.63, 0.955 |
| 5CT | 17.42, 7.05, 0.937 | 12.71, 7.59, 0.946 | 10.06, **4.96, 0.977** | 11.37, 6.59, 0.955 |
| 5DT | 17.65, 7.05, 0.935 | 12.31, 7.59, 0.947 | **9.98**, 5.02, 0.976 | 11.32, 6.62, 0.955 |
| 5ET | 17.15, 7.02, 0.935 | **12.23**, 7.59, 0.947 | 10.00, 5.02, 0.977 | 11.30, 6.58, 0.955 |
| 5FT | 17.75, 7.03, 0.935 | 12.39, **7.56, 0.947** | 10.31, 5.01, 0.977 | 11.58, 6.56, 0.955 |
| 5GT | 17.53, **7.01**, 0.935 | 12.29, **7.56, 0.947** | 10.31, 5.00, 0.977 | 11.57, 6.52, 0.955 |
| 5GT* | 17.39, 7.22, 0.934 | 12.46, 7.61, 0.946 | **9.98**, 5.01, 0.976 | **11.24**, 6.64, 0.955 |
| 5HT | 17.27, 7.12, 0.936 | 12.38, 7.63, 0.946 | 10.27, 4.98, 0.976 | 11.37, 6.57, 0.955 |
| | *FTIR-ECC at 18 km* | | | |
| 5AT | 17.22, 6.95, 0.954 | 16.21, 6.08, 0.965 | 14.27, 5.43, 0.982 | 15.23, 6.04, 0.974 |
| 5BT | 16.92, 7.08, 0.952 | 15.24, 6.06, 0.966 | 14.22, 5.44, 0.982 | 14.93, 6.05, 0.974 |
| 5CT | **16.30**, 7.10, 0.952 | **15.20**, 6.03, 0.966 | 14.22, 5.40, 0.982 | **14.77**, 6.00, 0.975 |
| 5DT | 17.76, 6.71, 0.957 | 15.82, 6.13, 0.964 | 14.52, 5.42, 0.982 | 15.11, 5.99, 0.974 |
| 5ET | 17.38, 6.68, 0.958 | 15.91, 6.10, 0.965 | 14.46, 5.38, 0.982 | 14.99, 5.92, 0.975 |
| 5FT | 17.81, **6.62, 0.958** | 15.91, 6.17, 0.964 | 14.12, 5.39, 0.983 | 14.90, 6.01, 0.975 |
| 5GT | 17.33, 6.63, 0.958 | 15.84, 6.16, 0.964 | 14.12, 5.35, 0.983 | 14.90, 5.95, 0.975 |
| 5GT* | 16.47, 6.70, 0.958 | 15.96, 6.06, 0.965 | 14.21, **5.31, 0.983** | 14.95, 5.90, 0.976 |
| 5HT | 16.95, 6.93, 0.955 | 16.17, 6.06, 0.965 | **14.08**, 5.41, 0.982 | 14.99, 6.00, 0.975 |
| | *FTIR-ECC at 29 km* | | | |
| 5AT | 9.84, 7.46, 0.640 | 10.22, 4.54, 0.712 | 10.38, 3.79, 0.846 | 10.29, 5.41, 0.741 |
| 5BT | 11.53, 8.08, 0.576 | 12.31, 4.84, 0.674 | 11.89, 3.79, 0.847 | 11.86, 5.69, 0.718 |
| 5CT | 11.42, 8.24, 0.571 | 12.29, 4.87, 0.672 | 11.89, 3.79, 0.848 | 11.88, 5.75, 0.716 |
| 5DT | 10.71, 4.14, 0.769 | 9.19, 3.40, 0.827 | 10.80, 3.01, 0.890 | 10.45, 3.50, 0.840 |
| 5ET | 10.63, 4.22, 0.768 | 9.19, 3.41, 0.827 | 10.80, 3.02, 0.890 | 10.48, 3.53, 0.839 |
| 5FT | 10.24, **3.69, 0.802** | **8.34, 2.60, 0.891** | 7.82, 2.91, 0.876 | 8.10, 3.26, 0.842 |
| 5GT | 10.05, 3.71, 0.791 | 8.37, 2.62, 0.890 | **7.80, 2.88, 0.878** | **8.05, 3.22, 0.842** |
| 5GT* | 10.68, 6.48, 0.722 | 8.74, 4.27, 0.744 | 8.07, 3.79, 0.835 | 8.55, 5.17, 0.752 |
| 5HT | 9.26, 7.56, 0.638 | 10.21, 4.53, 0.714 | 10.38, 3.78, 0.846 | 10.27, 5.44, 0.740 |





ture treatments: the bias is ∼16-18% up to the UTLS region and it drops up to ∼9-12% in the middle stratosphere for all
configurations. Nonetheless, for the IFS 120/5HR, together with an overall reduction of the bias (especially up to the UTLS
altitudes), the cross-interferences between the ILS, temperature, and $O_3$ concentrations become evident in the middle strato-
sphere. Considering the 2008-2018 period as reference (better instrumental alignment and more FTIR-ECC coincidences),
above the middle stratosphere the most accurate $O_3$ profiles are obtained with the most refined ILS set-ups with and without
temperature retrieval (5G/5GT, and the averaged set-ups 5G'/5GT') with bias ranging between ∼6-8%. At lower altitudes, the
differences between the strategies are not significant and lie within the overall variance.

    More remarkable discrepancies between the ILS approaches would be expected beyond the middle stratosphere, as suggested
by the uncertainty analysis (see Figure 2). In addition, compensations between the ILS, temperature, measurement errors, and
$O_3$ vertical distribution might occur at these altitudes to be consistent with the findings from the $O_3$ TC comparison (e.g.
the subtle improvement due to the PE fit considering the cell-derived MEA values). Unfortunately, the ECC database does
not allow these effects to be examined in detail given its limited altitude coverage. Other measurement techniques, such as
ground-based LIDARs or microwave radiometers capable of probing the $O_3$ profile at higher altitudes, could therefore provide
an added value for the $O_3$ vertical profile analysis.

## 6   Summary and Conclusions

A precise instrumental characterisation is indispensable to retrieve correct information from the measured solar absorption
spectra by ground-based FTIR (Fourier Transform Infrared) spectrometers, since it directly affects the absorption line shapes
of atmospheric gases. Currently, different approaches are used to deal with the FTIR instrumental response through the spec-
trometer's ILS (Instrumental Line Shape) function. The most widely-used options are to assume an ideal instrument, to retrieve
the instrumental information simultaneously with the atmospheric gas concentrations from the measured solar absorption
spectra, or to determine it from independent gas-cell measurements. In this context, this paper has assessed the impact of these
strategies on the quality of the FTIR ozone ($O_3$) products (total columns and vertical profiles) using the 20-year time series of
FTIR measurements acquired at the subtropical Izaña Observatory (IZO) between 1999 and 2018.

    The theoretical and experimental quality assessment addressed by this work has documented how critical the treatment of
the ILS function can be. Assuming an ideal ILS can provide imprecise $O_3$ concentrations, especially when the instrument is not
properly aligned, while retrieving the ILS information in the inversion procedure has proven to lead to a misinterpretation of the
actual $O_3$ variations on a daily and seasonal scale (the more sophisticated the ILS fit, the more pronounced this artifact). The
optimal approach to deal with the FTIR instrumental characterisation is therefore the continuous monitoring of the ILS function
by means of independent data, such as the low-pressure $N_2O$-cell measurements used in this study. These measurements should
be routinely taken about every two months for stable FTIR instruments, such as IFS 120/5HR, and even more frequently for
IFS 120M spectrometers.
Nonetheless, if the simultaneous ILS retrieval is carried out due to technical necessities of the FTIR station (e.g. in absence of
gas-cell measurements), an atmospheric temperature profile retrieval is strongly recommended to improve the precision of the


$O_3$ retrievals as well as to reduce the cross-interference between the atmospheric temperature and instrumental performance. However, this enhancement is only achieved provided the FTIR spectrometer is stable over time. For unstable instruments, the temperature retrieval exhibits a drastic negative impact on $O_3$ products even though the ILS fit is simultaneously performed.

In this sense, it is worth highlighting that retrieving only the ILS jointly to the $O_3$ concentrations (without a simultaneous temperature retrieval) might worsen the precision of the FTIR $O_3$ products, and it may be a better approach to assume an ideal ILS function.

   Other tentative approaches to assess the FTIR instrumental performance from atmospheric trace gas retrievals have been also evaluated in this study (e.g. $CO_2$ and HF). This analysis reveals that, in absence of independent ILS measurements,

the most promising option is indeed the retrieval of the ILS parameters jointly with $O_3$ concentrations (provided the ILS estimates are averaged annually to reduce seasonal effects). Nonetheless, it is important to be aware that the selection of the ILS characterisation approach (and the instrumental degradation) can significantly affect the determination of the $O_3$ vertical distributions. The impact is especially critical beyond the lower stratosphere, where most of the $O_3$ recovery is currently taking place and instrumental artefacts might lead to imprecise observational estimates on its evolution. Given that the projected $O_3$

trends are rather small, highest quality $O_3$ measurements are mandatory.





## Appendix A: Monitoring of the ILS function from $N_2O$-cell measurements: non-clamped Phase Error Retrieval

Figure A1 shows the ILS time series evaluated from the $N_2O$-cell measurements assuming a non-clamped PE retrieval (i.e. free PE) at ZPD, as described by Hase (2012). This figure furthermore includes the comparison between the cell-derived PE time series at an OPD of 133 cm for the clamped and non-clamped approaches, as well as the retrieved PE values from
the $O_3$ atmospheric retrievals using the set-up 5H, which superimposes a simultaneous fit of a PE offset to the retrieved path-dependent cell-derived PE. A remarkable discrepancy between the approaches is clearly observed, especially for the IFS 120M instrument. While the clamped PE retrievals from cell and solar spectra are rather consistent, the cell-derived non-clamped retrieval provides larger and more erratic PE corrections, which are in turn reproduced by the retrieved PE values from atmospheric spectra (set-up 5H). This fact seems to point to the $N_2O$-cell measurements may not contain enough information to
allow for a precise PE retrieval at ZPD. In addition, Table A1 summarises the comparison of the theoretical performance of the $O_3$ retrieval strategies using both cell-derived ILS estimates in terms of the obtained DOFS and fitting residuals. The clamped ILS approach consistently offer the best performance independently of the instrument's status/period: it provides a better vertical sensitivity and a superior interpretation of the measured atmospheric spectra. These results further corroborate that a clamped PE retrieval at ZPD when evaluating the $N_2O$-cell measurements is a superior choice to characterise the instrumental
performance of the NDACC FTIR spectrometers.

**Table A1.** Same as Table 2, but for the $O_3$ retrievals using the cell-derived ILS time series considering non-clamped PE retrievals at ZPD (Figure A1) (so-called $5X_{ncPE}$). For a better comparison the DOFS and fitting residuals obtained with the clamped cell-derived ILS time series have been also included (5A/5AT, and 5H/5HT).

| | DOFS | | | | Residuals (x$10^{-3}$) | | | |
| | 1999-2004 | 2005-2008 | 2008-2018 | 1999-2018 | 1999-2004 | 2005-2008 | 2008-2018 | 1999-2018 |
| Set-up | M, $\sigma$ | M, $\sigma$ | M, $\sigma$ | M, $\sigma$ | M, $\sigma$ | M, $\sigma$ | M, $\sigma$ | M, $\sigma$ |
|---|---|---|---|---|---|---|---|---|
| $5A_{ncPE}$ | 4.19, 0.29 | 4.46, 0.14 | 4.30, 0.11 | 4.31, 0.16 | 3.95, 2.04 | 2.79, 0.92 | 3.29, 0.59 | 3.28, 0.97 |
| $5H_{ncPE}$ | 4.35, 0.30 | 4.58, 0.15 | 4.51, 0.12 | 4.51, 0.17 | 3.34, 1.94 | 2.56, 0.88 | 2.72, 0.53 | 2.73, 0.90 |
| 5A | 4.29, 0.29 | 4.56, 0.15 | 4.52, 0.13 | 4.51, 0.18 | 3.51, 1.96 | 2.57, 0.86 | 2.70, 0.56 | 2.72, 0.93 |
| 5H | 4.35, 0.30 | 4.58, 0.15 | 4.52, 0.13 | 4.52, 0.17 | 3.37, 1.93 | 2.54, 0.85 | 2.70, 0.55 | 2.71, 0.90 |
| $5AT_{ncPE}$ | 3.99, 0.34 | 4.32, 0.19 | 4.14, 0.13 | 4.15, 0.19 | 3.90, 2.04 | 2.78, 0.91 | 3.28, 0.58 | 3.27, 0.97 |
| $5HT_{ncPE}$ | 4.13, 0.35 | 4.43, 0.21 | 4.35, 0.15 | 4.35, 0.21 | 3.27, 1.94 | 2.54, 0.87 | 2.70, 0.53 | 2.70, 0.89 |
| 5AT | 4.09, 0.34 | 4.42, 0.20 | 4.35, 0.15 | 4.35, 0.21 | 3.44, 1.96 | 2.55, 0.86 | 2.68, 0.56 | 2.70, 0.93 |
| 5HT | 4.14, 0.36 | 4.44, 0.20 | 4.36, 0.15 | 4.36, 0.21 | 3.28, 1.93 | 2.51, 0.85 | 2.67, 0.55 | 2.68, 0.90 |



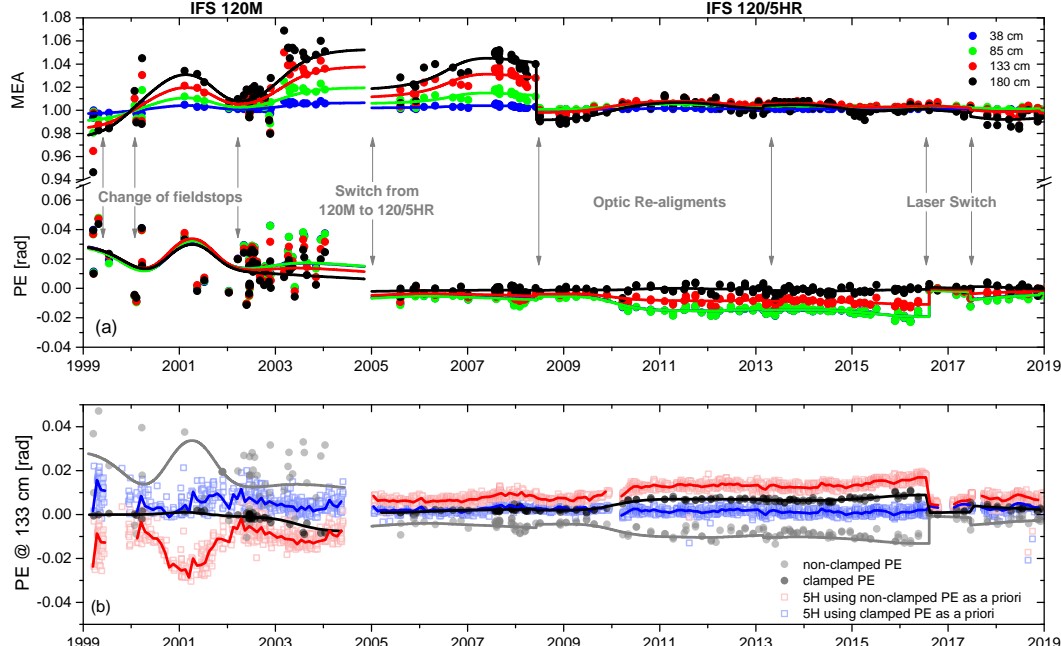

**Figure A1.** (a) Time series of the normalised MEA and PE values at four OPD (38, 85, 133 and 180 cm) between 1999 and 2018. Data points represent individual $N_2O$-cell measurements and solid lines depict the smoothed MEA and PE curves. The grey solid arrows indicate punctual interventions on the IZO FTIR instruments: changes of fieldstops between 1999 and 2004, switch from the IFS 120M to the IFS 120/5HR in January 2005, optic re-alignments in June 2008 and February 2013, and internal laser replacements in August 2016 and June 2017. (b) Time series of the PE values at an OPD of 133 cm, considering the non-clamped and clamped PE retrieval at ZPD, and the time series of the retrieved PE values from the $O_3$ retrievals using the 5H set-up and considering the non-clamped and clamped PE retrievals as a priori information. Note that the retrieved PE from the 5H set-ups is constant throughout the whole OPD range.

## Appendix B:  Monitoring of the ILS function from $CO_2$ and HF retrievals

An alternative approach to ensure the independence of the ILS and $O_3$ retrievals in case of ILS measurements are not available might be to retrieve the ILS information from atmospheric trace gas retrievals with well-known vertical distribution. For this purpose, the ILS parameters have been also evaluated from the measured absorption lines of very stable tropospheric and

stratospheric gases, such as $CO_2$ and HF, respectively. For the $CO_2$ approach, the four isolated $CO_2$ lines between 960-970 cm$^{-1}$ considered in the simultaneous temperature fit have been used, while the HF strategy retrieves the ILS parameters jointly with the HF concentrations from two micro-windows in 4000.90-4001.05 and 4038.85-4039.08 cm$^{-1}$ spectral range. Figure B1 displays the obtained ILS time series from the two approaches.

Both strategies are found to produce unrealistic ILS estimates, especially the HF approach, and a strong artificial annual cycle

in the retrieved MEA and PE values. The tropospheric $CO_2$ absorbing lines seem to be too broad to allow to accurately estimate the ILS, while the HF strategy is strongly affected by the mix between the ILS signatures and dynamical signals (seasonal shift





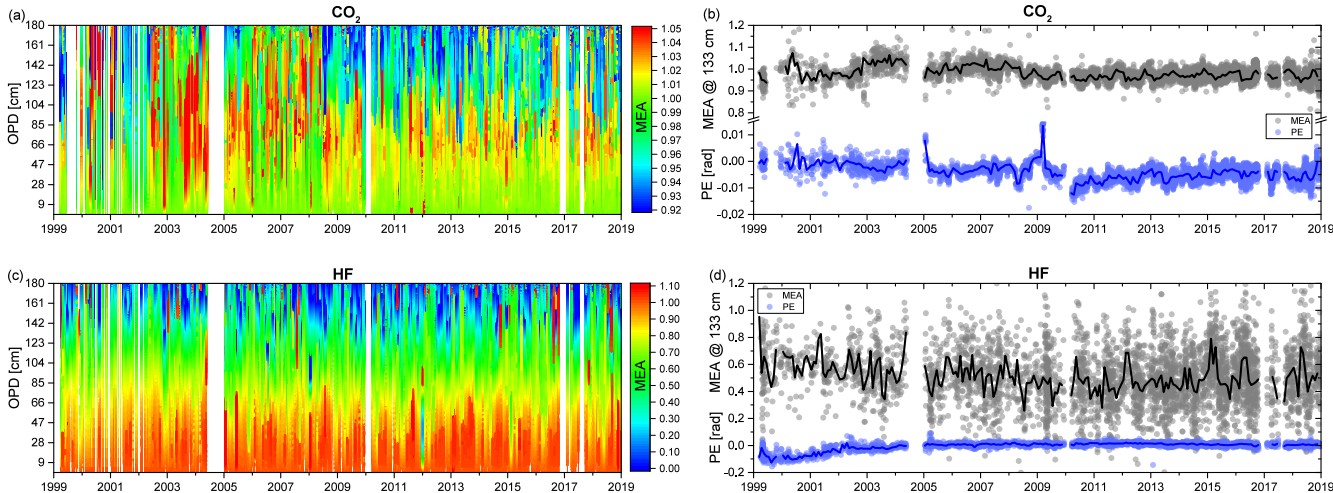

**Figure B1.** Time series of the MEA as a function of the OPD from 1999 to 2018 evaluated from simultaneous ILS retrievals (a) from $CO_2$, and (c) from HF absorbing lines. (b), and (d) same as (a), and (b), but for the MEA at an OPD of 133 cm and the phase error (PE), which is constant throughout the whole OPD range. Note that the coloured scale for the MEA is different for the sub-plots (a) and (c).

of the tropopause height, and stratosphere-troposphere exchange events). In the light of these results, in absence of independent ILS measurements, the most promising approach to characterise the instrumental performance might be to retrieve the ILS parameters jointly with $O_3$ concentrations (provided the ILS estimates are averaged annually to reduce seasonal effects).

*Data availability.* The FTIR and Brewer data are available by request to the corresponding authors, while the ozone sondes are available at the NDACC archive (www.ndaccdemo.org).

*Author contributions.* The manuscript was prepared by O.G. and E.S. with contributions from all co-authors. F.H. developed the LINEFIT and PROFFIT retrieval codes, and together with M.S. and T.B., discussed the results and participated in the retrievals analysis. O.G., M.S., and E.S. taken the routine FTIR measurements and regular cell measurements since 1999, and performed the maintenance and quality-control
of the FTIR instruments. A.R., S.F., and V.C. managed the Brewer spectrometers and elaborated on the ozone observations. Finally, C.T. and N.P. are in charge of the ozone sonde programme at IZO.

*Competing interests.* The authors declare no conflict of interest.



*Acknowledgements.* The Izaña FTIR station has been supported by the German Bundesministerium für Wirtschaft und Energie (BMWi) via DLRunder grants 50EE1711A and by the Helmholtz Association via the research programme ATMO. In addition, this research was

funded by the European Research Council under FP7/(2007–2013)/ERC grant agreement no. 256961 (project MUSICA), by the Deutsche Forschungsgemeinschaft for the project MOTIV (GeschaFTIRzeichen SCHN 1126/2-1), by the Ministerio de Economía y Competitividad from Spain through the projects CGL2012-37505 (project NOVIA) and CGL2016-80688-P (project INMENSE), and by EUMETSAT under its Fellowship Programme (project VALIASI).



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
