# Peer review of "Impact of instrumental line shape characterisation on ozone monitoring by FTIR spectrometry"

_Atmospheric Measurement Techniques, 2022_

## Author Comment (AC1)

**Comment on amt-2022-44 "Impact of instrumental line shape characterisation on ozone monitoring by FTIR spectrometry" by Omaira E. García et al.**

**Anonymous Referee #1**

Please find below the response to Referee #1's comments (in bold her/his comments and in italic the authors' replies).

General comments

**The IRWG of NDACC is an international network bringing together more than 20 observational FTIR sites over the world since the 1990s. In comparison to other observational methods used for the investigation of gaseous composition of the atmosphere, FTIR spectrometry provides a unique advantage of simultaneous measurements of the total columns (or even profiles) of a number of climatically active gases. The significant efforts are being made by the IRWG community to develop the unified retrieval strategies for deriving total columns/profiles of atmospheric species including O3. The main target of these efforts is verifying and harmonizing the results obtained by different groups operating FTIR sites. To get reliable information on O3 trends in the stratosphere which are currently nearly zero, it is necessary to provide the FTIR products of high accuracy and precision. Achieving this goal requires knowledge of the parameters characterizing the alignment of FTIR spectrometer (instrumental line shape function, ILS) and correct accounting of ILS in the retrieval procedure. Paper by García et al. is devoted to the detailed study of the influence of several ILS approaches (used in the procedure of inverse problem solving) on the O3 retrieval results (focusing the stratosphere). FTIR instruments having different alignment status are considered.**

**The manuscript corresponds to the AMT main subject areas and can be recommended for publication (minor revision is required) after a few points are addressed (please, see specific comments section)**

Specific comments

**1)      Results presented in Appendix B deserve to be moved to the main text of the paper as a separate section. But the final decision is up to the authors.**

*Appendix B aims to document how alternative approaches to evaluate the Instrumental Line Shape (ILS) function from atmospheric trace gas retrievals with well-known vertical distribution (i.e. carbon dioxide, $CO_2$, and hydrogen fluoride, HF), as suggested by previous works, are not precise enough to evaluate ozone ($O_3$) retrievals. Both strategies are found to produce unrealistic ILS estimates, therefore these approaches were not included in the main study. We would like to thank the referee for pointing this out. However, we would like to keep these findings in a separate Appendix to allow readers to be focused on the final ILS strategies presented and tested in this work.*

**2)       Authors tested several approaches of ILS for the getting best retrieval results on O3. Is there a "universal" recipe for the processing FTIR observations (for example, archive spectra) in the absence of information on instrument alignment (ILS function)? Is it correct that in the case of the ideal ILS function should be used for overall spectra processing? Is it possible to create a homogeneous O3 row by stitching separate O3 time series obtained as a result of processing FTIR spectra using different ILS approaches? Analysing such a complex time series can be an additional challenge to reveal long-term O3 trends close to zero.**

*One of the main motivation of this work is indeed that there is no a universal or standard approach to evaluate the ILS function of ground-based FTIR instruments. In this context, this study pretends to assess the impact of the existing ILS treatments within the NDACC FTIR community on the FTIR $O_3$ products, as an exemplary case.*

*Nevertheless, we believe that the provision of a "universal" optimized recipe would be difficult. The history of site operation as quality of the spectrometer, availability of cell measurements, etc. will in general suggest different specific strategies for achieving the best data product. Although not fully satisfactory from the viewpoint of network traceability, the construction of appropriate schemes will certainly involve considerable amount of operator knowledge, so even if spectra would be archived, important auxiliary information would probably not be forwarded and might be finally lost. Whether an ideal ILS should be assumed also needs to be decided by the site operator, as this will depend on specific factors.*

*The results of the current work point to the optimal approach to deal with the FTIR instrumental characterisation is the continuous monitoring of the ILS function by means of independent data, such as the low-pressure $N_2O$-cell measurements. Nonetheless, if independent information on the instrument alignment is not available, an intermediate approach could be the simultaneous ILS retrieval together with the atmospheric temperature profile fit. The combined (ILS, $O_3$ and temperature) approach is found to be superior with respect to assuming an ideal ILS function. It improves the precision of the FTIR $O_3$ retrievals as well as reduces the cross-interference between the atmospheric temperature and instrumental performance for the IFS 120/5HR spectrometers. For more unstable instruments, such as the IFS 120M, the temperature retrieval exhibits a drastic negative impact on $O_3$ products even though the ILS fit is simultaneously performed.*

*Therefore, the strategy of TCCON to use HCl cells in the solar beam for achieving a complete documentation of the ILS performance is a step ahead (NDACC might in future achieve a similar ILS monitoring at least for the InSb detector by using HBr cells).*

*Regarding the strategy of combining $O_3$ retrievals from different ILS approaches, we fully agree to the referee that this is a challenging task. Nevertheless, stitching the full time series together using different approaches might be unavoidable and without alternative, as we typically face technical progress on the instrumentation and ILS monitoring procedures over the years. It would be good to include such information on sub-periods in the metadata.*

**3)     It is not quite clear whether the AVKs (averaging kernels) were taken into account when comparing the O3 results obtained by the FTIR and Brewer techniques?**

*The Brewer technique only provides $O_3$ amounts in the integrated total column and, so far authors know, information about vertical sensitivity is not available. Therefore, the Brewer and FTIR observations are straightforwardly compared without taking the FTIR vertical sensitivity (i.e. retrieved averaging kernels) into account.*

---

## Author Comment (AC2)

**Comment on amt-2022-44 "Impact of instrumental line shape characterisation on ozone monitoring by FTIR spectrometry" by Omaira E. García et al.**

**Anonymous Referee #2**

Please find below the response to Referee #2's comments (in bold her/his comments and in italic the authors' replies).

**General comments**

**The work by Garcia et al. focus on the impact of the instrumenta line shape of the high resolution FTIR spectrometer on the retrieval of vertical ozone profiles. The authors evaluate ver systematically more than 8 retrieval steregies to use y estimate the instrumental lineshape and evaluate these strategies using the long term measurements of FTIR spectra together with measurements from Brewer and the frequent ozone soundings. These collocated measurements at the Izaña observatorio are an unique posibility to evaluate the different retrieval strategies with respect to the instrumenta line shape and to study the sensitivity of the ozone profile retrieval on the estimation of the ILS.**

**The optimization of retrieval strategies for vertical gas profiles and especially of ozone by a very focused study is going to improve this remote sensing method towards a more exact measuremet method which is an important step for measurements in the NDACC network. The work fits perfectly in the scope of AMT, the manuscript is well written and clear presented.**

**In my opinion the manuscript is ready and paper should be published as is.**

**Specific comments**

**1) I would recomend to include a headline like "relative difference" in table 1,3,4 and it would be helpfull to see in addition to the infromation with respect to the brewer column measurements, a relative diference between the retrieval steragies, maybe with respect to the favourite retrieval steragies of the authors. This would ilustrate clear how sensible the ozone profile depend on the retrieval sterategy with respect to the ILS.**

*Following the Referee's suggestion, the comparison between the retrieved $O_3$ products (total columns and profiles) from the ILS retrieval strategies has been included in Section 4. To do so, the following text, table and figure has been added.*

*The differences between the ILS treatments are transferred to the $O_3$ TCs and profiles as summarised in Table 1 and Figure 1, respectively. The set-ups not retrieving MEA information provides the largest bias with respect to the cell-derived ILS $O_3$ TCs (i.e. up to 0.3% for 5B/5C in the 2005-May 2008 period), whereas the most significant variability*

is observed for the most refined ILS set-ups (i.e. up to 0.7% for 5F/5G in the 2005-May 2008 period). As expected, the simultaneous temperature retrieval strongly affects the differences between the ILS treatments due to the cross-interference between the ILS, and the $O_3$ and temperature profiles (especially beyond the lower stratosphere, as illustrated in Figure 1).

| | 1999-2004 | 2005-2008 | 2008-2018 | 1999-2018 |
|---|---|---|---|---|
| Set-up | M[%], $\sigma$[%], R | M[%], $\sigma$[%],R | M[%], $\sigma$[%], R | M[%], $\sigma$[%], R |
| 5B | 0.13, 0.12, 1.000 | 0.27, 0.07, 1.000 | 0.11, 0.04, 1.000 | 0.12, 0.08, 1.000 |
| 5C | 0.10, 0.13, 1.000 | 0.27, 0.07, 1.000 | 0.10, 0.04, 1.000 | 0.12, 0.08, 1.000 |
| 5D | -0.05, 0.47, 0.998 | -0.01, 0.57, 0.994 | 0.01, 0.32, 0.999 | 0.01, 0.38, 0.998 |
| 5E | -0.08, 0.47, 0.998 | -0.02, 0.59, 0.993 | 0.01, 0.32, 0.999 | 0.00, 0.38, 0.998 |
| 5F | 0.00, 0.53, 0.998 | -0.08, 0.70, 0.991 | -0.16, 0.64, 0.993 | -0.15, 0.64, 0.994 |
| 5G | -0.01, 0.53, 0.998 | -0.08, 0.69, 0.991 | -0.16, 0.64, 0.993 | -0.15, 0.64, 0.993 |
| 5H | -0.06, 0.05, 1.000 | -0.01, 0.01, 1.000 | 0.00, 0.01, 1.000 | 0.00, 0.03, 1.000 |
| 5BT | 0.65, 0.59, 0.997 | 1.62, 0.39, 0.998 | 0.36, 0.17, 1.000 | 0.36, 0.17, 0.997 |
| 5CT | 0.63, 0.62, 0.997 | 1.63, 0.40, 0.998 | 0.36, 0.17, 1.000 | 0.42, 0.49, 0.997 |
| 5DT | 0.30, 1.88, 0.976 | 0.29, 1.82, 0.940 | 0.11, 0.85, 0.991 | 0.14, 1.16, 0.982 |
| 5ET | 0.24, 1.87, 0.976 | 0.30, 1.74, 0.942 | 0.11, 0.89, 0.990 | 0.14, 1.17, 0.982 |
| 5FT | 0.30, 1.87, 0.976 | 0.33, 1.65, 0.950 | 0.25, 1.09, 0.983 | 0.26, 1.28, 0.977 |
| 5GT | 0.23, 1.85, 0.976 | 0.34, 1.76, 0.943 | 0.25, 1.10, 0.983 | 0.26, 1.30, 0.977 |
| 5HT | -0.18, 0.25, 1.000 | 0.00, 0.02, 1.000 | 0.02, 0.03, 1.000 | 0.01, 0.09, 1.000 |

Table 1. Summary of statistics for the $O_3$ TC comparison for the set-ups 5B/5BT, 5C/5CT, 5D/5DT, 5E/5ET, 5F/5FT, 5G/5GT, and 5H/5HT with respect to 5A/5AT: median (M, in %) and standard deviation ($\sigma$, in %) of the relative differences (RD, 5X/5XT - 5A/5AT), and Pearson correlation coefficient (R) for the periods 1999-2004, 2005-May 2008 and June 2008-2018, and for the entire time series (1999-2018). The number of quality-filtered measurements is 466, 683, and 3775 for the three periods, respectively, and 4924 for the whole dataset.

[Figure]

Figure 1. Summary of the $O_3$ profile comparison for the set-ups 5B/5BT, and 5G/5GT with respect to 5A/5AT for the periods 1999-2004, 2005-May 2008 and June 2008-2018, and for the entire time series (1999-2018). (a), (c), and (e) display the vertical profiles of the median (M) RD (5X/5XT - 5A/5AT, in %) for the three periods, respectively. (b), (d), and (f) same as (a), (c), and (e), but for the standard deviation of RD distributions ($\sigma$, in %).

*Regarding including the headline of "relative difference" in the tables that summarise the total columns and profiles comparisons, this information is already mentioned in the table captions. In addition, these tables include information not only referred to the relative differences (i.e. Pearson correlation values). Therefore, we would like to keep the table captions as they are to avoid readears to get confused with the information contained.*

**2)     I would recomend to include a little more basic information about the modulation eficiency and the phase error in the begining of the section 3.1, so that the reader get the important information which and how the ILS depends on both parameters without consulting the cited papers.**

*Following the Referee's suggestion, the following information has been added to Section 3.1:*

*The ILS function is the Fourier transform of the weighting applied to the interferogram (Davis et al., 2001). In the case of ideal instruments, the ILS is affected only by modulation loss that is due to the self-apodization of the interferometer, accepting a finite field of view, and is symmetric (Hase et al., 1999). For real instruments, the ILS also accounts for misalignments and optical aberrations of the spectrometer and is equivalent to a complex modulation efficiency (ME) in the interferogram. The phase corrected interferogram generated by a spectral line is of the form (Hase et al., 1999):*

$$IFG(\delta) \sim MEA(\delta) \cdot cos(2\pi\sigma - PE(\delta))$$

*where IFG($\delta$) is the interferogram (with $\delta$ as the mirror displacement), $\sigma$ is the wavenumber, MEA is the modulation efficiency amplitude, and PE is the modulation efficiency phase error.*

*The MEA and PE are parameters describing the deviations from the expected nominal ILS. Because the measurement process of an FTIR spectrometer is performed in the interferogram domain, this parameterisation refers to the interferogram, LINEFIT uses 20 equidistant grid points up to OPDmax and assumes a smooth variation of MEA and PE along OPD. MEA (normalized to unity at zero path difference, ZPD) is a measure of ILS width, and a decline of MEA towards indicates a broader ILS, while a rise indicates a narrower ILS with stronger sidelobes. A curving PE indicates ILS asymmetry (while a linear rise is equivalent to a spectral shift of the ILS, but does not a distortion of its shape). From the physical viewpoint, interferometric misalignments, deviations from the aspired circular interferometric field-of-view, OPD-dependent vignetting effects, and mismatch between the wavefronts of the reference laser wavefront and the infrared beam are main drivers of ILS imperfections.*

*Davis, S.P.; Abrams, M.C.; Brault, J.W. Fourier Transform Spectrometry; Academic Press: Cambridge, MA, USA, ISBN 0-12-042510-6, 2001.*

**3)      It is very interessting that the authors recomend to fit the pase error at ZPD, but use the modulation efficiemncy from the cell measurements. It would be very nice if the authors might try to give a possible physical explanation, why a fit of the "pase error" at ZPD from an individual spectrum imporves the retrieval? Does it depend on alignment the temperature of the beamsplitter the phasecorrection duringt the calculation of the spectrum?**

*This is an interesting question. Ideally, the phase correction should guarantee zero PE at ZPD. However, this phase information is effectively deduced from a rather small number of interferogram points in the centerburst, so there is noise superimposed on the phase spectrum. This problem has been highlighted by the late Luc Delbouille, who suggested to do repeated measurements of the centerburst between the full-resolution scans and use this low-noise phase spectrum for the phase correction [L. Delbouille, priv. comm.]. So far, such schemes have not been realized for operational NDACC work. Besides, as the referee suggests, we cannot rule out the existence of other effects distorting the phase spectrum, as the laser does not use the full beam diameter of the infrared beam and the resulting wavefront errors might depend on temperature (e.g. slight deformations of the cubecorner mirrors).*

---

## Author Response (AR1)

Dear Editor,

All comments and suggestions from both Referees have been assessed and included in the revised manuscript.

In order to be consistent with the related ozone AMT paper, entitled "Improved ozone monitoring by ground-based FTIR spectrometry" (AMT - Improved ozone monitoring by ground-based FTIR spectrometry (copernicus.org), the FTIR and ECC comparison at representative altitudes (Table 5 of the preprint) has been replaced by the comparison between ozone partial columns using the altitude levels as defined in García et al. (2012), i.e., the layers that are sufficiently well-detectable by the ground-based FTIR system (2.37-13 km, 12.-23 km and 22-29 km). The content of Section 5.2 has been accordingly adapted, but we would like to remark that significant changes has not been introduced in the discussion.

Many thanks and best regards,

Omaira García et al.